# Inferring ongoing cancer evolution from single tumour biopsies using synthetic supervised learning

**Tom W. Ouellette**[1,2]*, **Philip Awadalla**[1,2]*

**1** Ontario Institute for Cancer Research, Department of Computational Biology, Toronto, Ontario, Canada, **2** Department of Molecular Genetics, Temerty Faculty of Medicine, University of Toronto, Toronto, Ontario, Canada

\* tom.ouellette@oicr.on.ca (TWO); or philip.awadalla@oicr.on.ca (PA)

**Data Availability Statement:** All TumE predictions in synthetic and empirical datasets, intermediate processing data, data used for generating figures, and fully trained deep learning models can be found at the Zenodo repository https://doi.org/10.

## Abstract

Variant allele frequencies (VAF) encode ongoing evolution and subclonal selection in growing tumours. However, existing methods that utilize VAF information for cancer evolutionary inference are compressive, slow, or incorrectly specify the underlying cancer evolutionary dynamics. Here, we provide a proof-of-principle synthetic supervised learning method, TumE, that integrates simulated models of cancer evolution with Bayesian neural networks, to infer ongoing selection in bulk-sequenced single tumour biopsies. Analyses in synthetic and patient tumours show that TumE significantly improves both accuracy and inference time per sample when detecting positive selection, deconvoluting selected subclonal populations, and estimating subclone frequency. Importantly, we show how transfer learning can leverage stored knowledge within TumE models for related evolutionary inference tasks—substantially reducing data and computational time for further model development and providing a library of recyclable deep learning models for the cancer evolution community. This extensible framework provides a foundation and future directions for harnessing progressive computational methods for the benefit of cancer genomics and, in turn, the cancer patient.

## Author summary

Recent pioneering work modeling the evolutionary dynamics in growing tumours has provided fundamental insight into the quantitative relationship between mutation frequency and evolutionary parameters. However, existing approaches to infer cancer evolutionary parameters from bulk-sequenced tumour biopsies suffer from method-specific limitations. Individual summary statistics compress data into a single value with potentially limiting assumptions, approximate Bayesian computation suffers from the curse of dimensionality and can take hours per sample for making estimates, and mixture models are only implicitly connected to the evolutionary model. In this study, we show how we can combine existing theory, cancer evolution simulations, and deep learning to make estimates of cancer evolutionary parameters in single tumour biopsies. Our method allows us to avoid information loss that comes with compressing data into a single statistic prior

5281/zenodo.5931436. Whole-genome sequenced AML samples were retrieved from Griffith et al [31]. Multi-region whole-exome sequenced mismatch deficient repair gastro-oesophageal samples were retrieved from von Loga et al [33]. The remaining whole-genome sequenced samples were retrieved from PCAWG [11] at https://dcc.icgc.org/. We provide hosting of the interactive electronic supplementary at https://tomouellette.gitlab.io/ouellette_awadalla_2021/. Scripts for generating figures and analyses can be found at https://doi.org/10.5281/zenodo.5931436. Code for generating synthetic tumour sequencing data can be found at https://github.com/tomouellette/CanEvolve.jl. Code for performing inference with TumE can be found at https://github.com/tomouellette/TumE.

**Funding:** This work was supported by the Ontario Ministry of Research and Innovation award to PA. TWO was supported by a Canadian Institutes of Health Research (CIHR) Frederick Banting and Charles Best Canada Graduate Scholarship. The funders had no role in study design, data collection and analysis, decision to publish, or preparation of the manuscript.

to simulation, to separate simulation and training from prediction which ultimately makes inferences faster (~1 second), and to extend existing models with transfer learning when evolutionary assumptions are modified. Fast, efficient analysis of high-coverage, high-quality whole-genome and exome sequenced tumour biopsies reveals evolutionary selection patterns over or under-detected relative to alternative approaches. Our framework provides a working example and multiple new avenues of research for how to integrate simulations with deep learning for evolutionary inference.

## Introduction

Cancer is a disease characterized by unrelenting tissue growth and clonal evolution. During evolution, genetic and epigenetic aberrations provide the reservoir for dysfunctional cellular phenotypes that maintain a tumour's replicative advantage, while, over time, fluctuating physiological and ecological properties within the tumour microenvironment drive the need for updated adaptations that sustain immortality [1]. Overall, the complex interplay between mutation accumulation and microenvironmental changes leads to a high degree of both cellular and genetic heterogeneity and, by proxy, composite subclonal structure in tumours [2–4]. Naturally, the desire to better understand the evolutionary and subclonal dynamics in growing tumour populations has become a major task for cancer genomics—with goals of forecasting tumour progression, developing adaptive evolutionary therapies, and deconvoluting the genetic architecture that drives adaptation [3,5–8].

However, a significant hurdle in understanding cancer evolution *in vivo* are the clinical constraints surrounding serial sequencing, through space or time. For this reason, tumour biopsies are primarily sequenced in bulk from a single site and at a single time point. Although multi-region and single-cell data are becoming increasingly utilized, single time point, bulk sequenced biopsies still represent the major accessible data source for precision genomics guided treatment [9] and for studying cancer genomics and evolution in patients [10,11]. Given this limitation, a reasonable strategy for inferring evolution in single tumor biopsies has been to utilize theoretical population genetics to capture signatures of selection from the variant allele frequency (VAF) distribution [7,12–17]. The premise being that fitness-altering mutations will deterministically change in frequency over time, leading to characteristic and quantifiable deviations in the VAF distribution relative to some neutral evolutionary scenario [18].

VAF-based methods have been employed to differentiate between positive selection and neutral evolution [12,13], to examine growth patterns [19], to quantify subclonal fitness and time subclonal emergence [7,15], and to build population genetics informed mixture models [16] that account for neutral dynamics, that shape, to some extent, all tumour populations. With that said, existing VAF-based methods used to infer cancer evolution, although mechanistic and useful, have apparent limitations. For example, single statistics [12,20,21] are maximally compressive and cannot infer complex information, approximate Bayesian computation methods suffer from the curse of dimensionality and can be prohibitively slow due to a rate-limiting simulation step required for each sample [7,22,23], and mixture models, used to identify subclonal populations [16,24,25], are only implicitly connected to an underlying model of evolution and, until recently [16], have been built under incorrect assumptions that have led to systematic overestimation in the number of subclonal populations in sequenced tumours.

To address these limitations, we contribute a proof-of-principle synthetic supervised deep learning approach, TumE, for quantifying and classifying the evolutionary and subclonal

dynamics in bulk sequenced tumours biopsies using purity-corrected variant allele frequency (VAF) information from diploid genomic regions. By generating synthetic VAF distributions, as a proxy for evolutionary ground truth, from plausible simulations of tumour evolution, we were able to build inference models that accurately classify and quantify evolutionary (e.g. positive selection versus neutral evolution) and subclonal dynamics (e.g. subclone frequency) in real patient tumours while capturing uncertainty in our estimates, via a form of approximate Bayesian inference called Monte Carlo dropout [26,27]. Importantly, our method further highlights the power of utilizing deep learning for inference—namely exploiting stored knowledge via transfer learning. By recycling our models for new evolutionary prediction tasks, we reduce the computational burden associated with the generation of synthetic or simulated data. We validated our synthetic supervised learning approach in millions of synthetic tumours and applied TumE to 88 copy-number and purity corrected whole-genome (WGS) and whole-exome (WES) sequenced tumour biopsies.

## Results

### Inferring cancer evolution using synthetic supervised deep learning

Synthetic supervised, or simulation-based, deep learning has been shown to be equivalent to amortized approximate inference under a generative model [28]. Therefore, by optimizing a neural network using realistic synthetic data **x** generated from a stochastic generative process $p$(**x**,**z**|**θ**), where **θ** indicates the prior or parameters that define the simulation and **z** indicates the latent variables generated during simulation, we can build inference models that approximate our true posterior of interest $p$(**θ**,**z**|**x**). In our case, by optimizing a neural network using synthetic VAF distributions sampled from $p$(**x**,**z**|**θ**), we can build models for evolutionary inference in sequenced tumour biopsies (Fig 1 and Methods).

To generate synthetic data that properly captured evolutionary dynamics in patient tumours, we implemented a simulation framework, i.e. a stochastic generative process $p$(**x**,**z**|**θ**), combining two complementary approaches to improve the speed and efficiency of

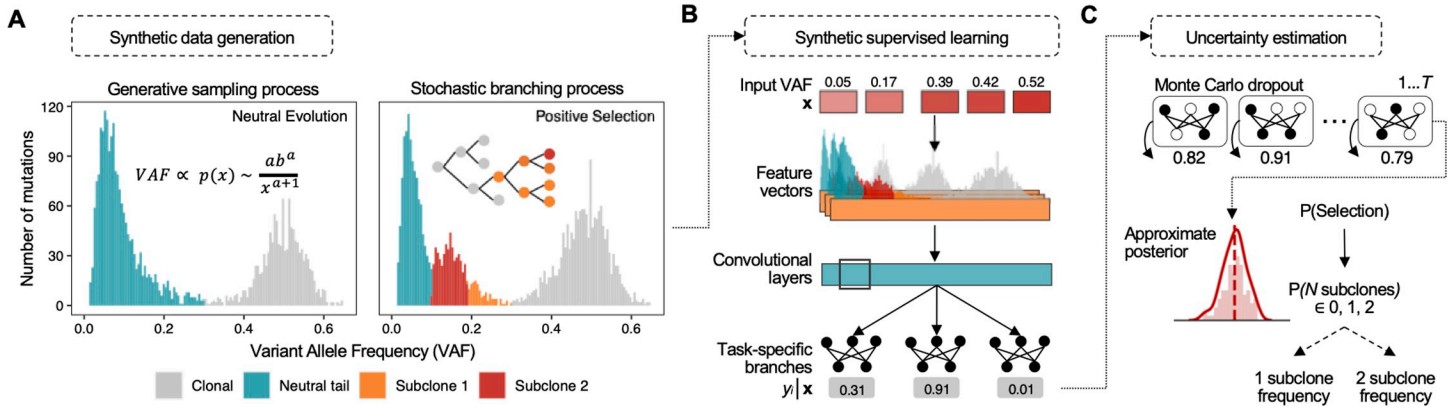

**Fig 1. (A)** TumE integrates a generative sampling process and stochastic simulation of cancer evolution to build well-specified synthetic variant allele frequency (VAF) distributions with respect to data observed in bulk sequenced tumour biopsies. Assuming copy-neutral diploid regions of tumour genomes, the generative sampling process uses the observation that neutral VAF distributions can be described by a power-law or Pareto neutral 'tail' [16,30] in addition to a dispersed clonal peak. By sampling empirically valid Pareto distributions, rapid realizations of the null hypothesis of neutral evolution encoded in the VAF distribution can be created (Methods). The stochastic branching process model of tumour evolution is then used to link parameters and latent states, relevant to positive subclonal selection, back to VAF distributions (Methods). **(B)** Synthetic supervised learning utilizes neural networks capable of handling the complete dimensionality of the simulated VAF distributions, **x**, to solve the inverse problem of identifying the evolutionary parameters and latent states, **y**, assigned to each synthetic VAF distribution. **(C)** We can then quantify model uncertainty using a computationally efficient form of Bayesian deep learning called Monte Carlo dropout [26,26]. Approximate posteriors are generated by performing $T$ stochastic passes through the trained neural network.

synthetic data generation—one for tumours subject to positive selection and one for tumours evolving neutrally (Fig 1A).

For growing tumours simulated with positive selection, we utilized a well-established framework of cancer evolution that models exponential tumour growth under a stochastic branching process [7,12,13,15,19,29] and coupled this with a virtual biopsy procedure to account for sequencing noise observed in real patient tumours (adapted from [7]). In our model, we allowed for a completely stochastic arrival of driver mutations that multiplicatively increased the fitness of mutated subclones, and tracked the frequency of each subclone until the time of virtual biopsy (Methods). In this study, we define a subclone as a subpopulation of cells with a fitness or growth rate advantage relative to the background population (Methods) and consider subclones detectable if they are between ~10–40% VAF (20–80% cellular fraction).

For tumours that lacked selected subclones (neutrally evolving), we implemented a generative sampling process based on the observation that VAF distributions from tumours without positively selected subclones can be described by a power-law or Pareto distribution [16,30] in conjunction with a dispersed clonal peak (S1 Fig). Concisely, this process involved i) sampling allele frequencies from empirically realistic Pareto distributions to generate the neutral power-law 'tail' in the VAF distribution, ii) adding additional diploid clonal heterozygous mutations at 50% VAF, and then iii) injecting additional sequencing noise under a beta-binomial model (Methods). In general, a complete VAF distribution indicative of positive selection, and computed from heterozygous diploid mutations, includes a neutral power-law tail [12,16], a heterozygous clonal peak centered at ~50% VAF, and additional subclonal peak(s) in the intermediate frequency ranges (~10–40% VAF); whereas a neutrally evolving tumour, or one with undetectable selected subpopulations, lacks the characteristic subclonal peak(s) (Fig 1A). To ensure positively selected and neutrally evolving synthetic tumours were not out of distribution with each other given the alternate data generation approaches, we simulated synthetic tumours in pairs, assigning the neutral VAF distributions with equivalent parameters and mutations with respect to the paired positive selection simulation (Methods; pseudo algorithms S1 Text and examples S1 and S2 Figs).

Using this framework, we generated approximately 40 million synthetic tumours across varying mutation rates, selection coefficients, and sequencing noise parameters. We selected broad simulation parameter ranges that were consistent with previous computational studies and empirically estimated values (Methods and S1 Table). By generating synthetic tumours using well-specified simulations (comparison of real and synthetic data outlined in Methods and S3–S5 Figs), we were able to explicitly link each VAF distribution to the parameters and latent states that defined the underlying subclonal and evolutionary dynamics. We then used the millions of annotated synthetic VAF distributions to train hundreds of neural networks using a random hyperparameter search to make inferences on the evolutionary mode (positive selection or neutral evolution), the number of subclones (0, 1, or 2), and the subclone frequency at borderline to optimal sequencing depths (50–250X) for evolutionary analysis in cancer genomics (Fig 1B and Methods). To capture model-based uncertainty in our estimates, we implemented a form of Bayesian approximation for deep learning called Monte Carlo (MC) dropout [26,27] (Fig 1C and Methods). We used MC dropout to mitigate overconfident estimates in cases of high uncertainty or broad approximate posteriors. In general, we structured both neural network training and prediction to favour the more parsimonious explanation of the data (fewer subclones and neutral evolution; Methods). We show how using a classification threshold based on a lower bound of the MC dropout approximate posterior helps mitigate model overconfidence across changing subclone mutational burdens and frequencies in S6 Fig. Following training, we selected the top scoring models, for predicting the evolutionary mode, number of subclones, and subclone frequency, for further validation (Methods).

We outline the full synthetic supervised learning pipeline in Methods. In addition, we highlight that even though we model VAF distributions in patient tumours using point mutations from diploid regions, mutations in our framework, as with previous approaches [7,13,16], are agnostic to the underlying functional alteration, e.g. missense, silent, driver or copy number driving selection in patient tumours. This is because genome-wide linkage, a by-product of zero recombination, results in hitchhiking of any additional point mutations on the genetic background of any selected clone [3,18].

## Comparison of synthetic supervised learning to existing methods

To evaluate TumE performance on inferred estimates of selection, number of subclones, and subclone frequency, we simulated an additional ~2.8 million synthetic tumours under neutral evolution (0 subclones) and positive selection (1 or 2 detectable selected subclones) assessing the impact of variable sequencing depths (50–250x coverage) and read count overdispersion (0–0.3 rho) (Methods). We first compared TumE against frequency-based summary statistic approaches for differentiating between neutral evolution and positive selection and found that TumE significantly outperforms recently developed VAF summary statistics [12] (two-sided Wilcoxon test, $p = 2.7 \times 10^{-12}$) as well as common population genetic summary statistics [20,21] (two-sided Wilcoxon test, $p = 1.9 \times 10^{-8}$), based on AUROC (Fig 2A). Further, TumE outperforms each statistic individually when compared across all sequencing depth and overdispersion combinations analyzed here (ROC analysis; S7 Fig).

We next compared TumE against the only mixture model approach, MOBSTER [16], that explicitly and correctly considers the neutral dynamics within sequenced tumour VAF distributions to detect subclones. We found that TumE provides comparable or improved performance for predicting the number of subclones (precision-recall, S8 Fig) and for predicting subclone frequency (Fig 2B; correlation and mean absolute percentage error, S9 Fig) across all empirically relevant depth (50–250x coverage) and read count overdispersion (0–0.003 rho) combinations. However, as expected, we found that the performance of TumE and MOBSTER both degrade as sequencing depth decreases ($\leq$ 75x coverage) and overdispersion increases ($\geq$ 0.01) under a beta-binomial sequencing noise model (S8 and S9 Figs). Furthermore, additional analysis of subclone frequency estimates in the 2 subclone setting revealed that as inter-subclone distance increases, i.e., overlap of subclonal peaks decrease, the mean percentage error for predicting the frequency of both the lowest and highest frequency subclone decreases towards zero (S10–S12 Figs).

Given our simulation framework was based on certain approximating assumptions to improve computational speed and efficiency (namely small population size and no cell death; outlined in Methods), we sought to perform additional validation of evolutionary estimates in an alternative dataset of synthetic tumours [16]. The orthogonal dataset, described in Caravagna et al. 2020 [16], consisted of 150 synthetic tumours, 40 effectively neutral and 110 with one detectable subclone (between 10–45% VAF), sequenced to 120x depth and grown to a population size of $>10^8$ cells at birth rate of 1 and death rate of 0.2. To frame our predictions relative to existing methods, we applied TumE, MOBSTER, and a variational Bayesian mixture model sciClone [24] to the synthetic dataset. To make comparisons fair, we limited the maximum number of subclonal cluster assignments to 2 for both MOBSTER and sciClone, as this was the upper bound on TumE estimates (Methods). For sciClone, this meant setting the maximum number of mixture components to 4 (neutral tail, 2 subclones, and a clonal peak) as sciClone doesn't properly account for neutral dynamics (Pareto tail) observed in sequenced tumour populations. Both TumE and MOBSTER consistently identified the correct number of detectable subclones in the majority of cases while sciClone systematically overestimated the

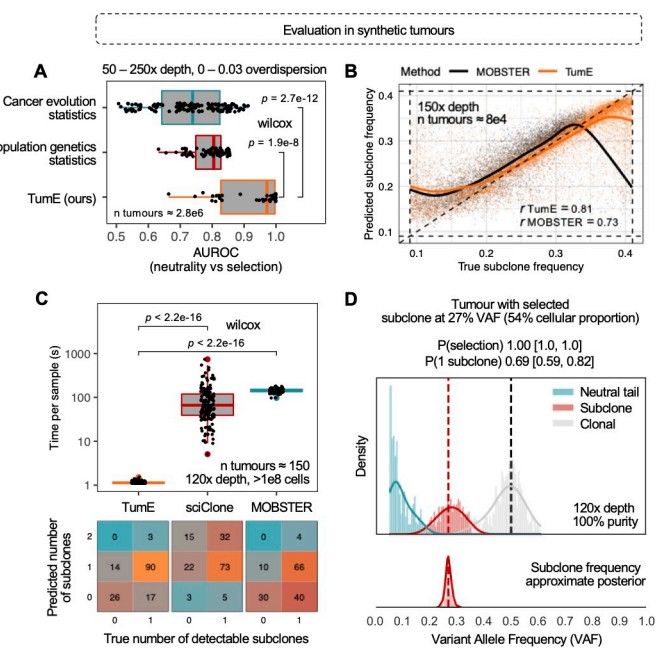

**Fig 2.** **(A)** In a cohort of 2.8 million synthetic tumours, TumE outperformed all existing common population genetic [20,21] and cancer evolution [7,12] specific summary statistics when differentiating between positive selection and neutral evolution, based on AUROC (two-sided Wilcoxon test). **(B)** Further, for predicting the true frequency of selected subclones, TumE provides comparable or better performance relative to the current state-of-the-art mixture model MOBSTER [16] that properly accounts for neutral dynamics in tumour populations. The panel shows correlation between the true and predicted subclone frequency in 80,000 synthetic tumours sequenced at 150x mean sequencing depth. **(C)** In an orthogonal dataset of 150 synthetic tumours [16] with either 0 or 1 detectable subclones, TumE was significantly faster at estimating the number of subclones (two-sided Wilcoxon test) than existing mixture model based methods sciClone [24] and MOBSTER [16] (measured in inference time per sample). In addition, only TumE and MOBSTER consistently identified the correct number of subclones, as both methods directly account for the neutral dynamics observed in tumour populations. **(D)** TumE estimates in a synthetic tumour sequenced at 120x mean sequencing depth and a subclone at 54% cellular fraction.

number of subclones, even after correcting estimates for the clonal peak and neutral tail (Fig 2C). However, relative to MOBSTER, which tended to converge to more parsimonious explanations of the data (i.e., 0 subclones), TumE was able to detect subclones at a higher rate (Fig 2C). In addition, compared to both sciClone and MOBSTER, TumE provided orders-of-magnitude faster estimates (two-sided Wilcoxon test, $p < 2.2 \times 10^{-16}$, Fig 2C), reducing run times per sample from minutes to ~1 second. We provide individual estimates with TumE for each of the 150 synthetic tumours, and an additional 750 synthetic tumours of variable sequencing depth from [16], in S13 and S14 Figs. We provide an example TumE output for a synthetic tumour with a single detectable subclone in Fig 2D.

In this study, we note that the birth and death rate were set to fixed values (birth rate = log (2), death rate = 0, in line with [7]) to additionally improve the computational efficiency of the stochastic simulations of positively selected tumour populations. Therefore, an additional factor that may impact the accurate detection of selection and subclones with TumE is variable birth and death rates in growing tumours. For example, an elevated cell death can lead to an increase in the number of passenger mutations that are swept to higher frequencies during subclonal selection. In regard to the VAF distribution, this elevated number of mutations 'trailing' the subclonal peak may obscure lower frequency subclones or, alternatively, lead to spurious identification of additional subclones due to an elevated number of neutral mutations entering the subclonal frequency range. To assess the impact of variable growth rates, we

generated an additional 6 million synthetic tumours across 26 different birth and death rate combinations (simulation parameters outlined in S1 Table). Overall, we find that our estimates are robust to changes in tumour growth rates. Any errors that do occur only appear to increase the number of parsimonious explanations of the data (e.g., classifying 2 subclones as 1; S15 Fig). In addition, the prediction of subclone frequency also remained consistent across all the birth and death rate combinations evaluated here (S16 Fig).

One further consideration is that all models in this study were trained on synthetic data emulating VAF distributions generated from heterozygous diploid mutations collected from 100% pure tumours. Given that TumE requires purity corrected VAF information from diploid regions, tumour impurity has the potential to confound estimates by presenting a spurious number of mutations within subclonal frequency ranges. Although VAFs can be easily corrected for impurity in diploid regions (i.e., diploid VAF / purity), original tumour purity estimates may also be incorrect. To ensure incorrect purity estimates did not confound our predictions, we implemented a peak-finding method that adjusts the VAF distribution to a theoretical 100% purity state (Methods). Using this adjustment approach, analyses in simulated datasets showed that TumE estimates can maintain a false positive rate for positive selection of less than ~1–5% in samples with up to 25% error in purity estimates and at minimum effective sequencing depths (purity * sequencing depth) as low as ~40x (S17 and S18 Figs).

## Analysis of whole-genome and exome sequenced tumour biopsies

To make the utility of synthetic supervised learning concrete, we first evaluated TumE in 'gold-standard' tumour biopsies commonly used to evaluate mixture model based approaches, namely a deep sequenced (~320x coverage, 90.7% purity) acute myeloid leukemia (AML) sample from Griffith et al. [31] and a deep sequenced (~226x coverage, 71.2% purity) breast adenocarcinoma sample retrieved from the pan-cancer analysis of whole genomes (PCAWG) [11] but originally from [32]. In both cases, we recovered the correct evolutionary mode, number of subclones, and subclone frequencies (Fig 3A and 3B). In addition, because we provide accurate subclone frequency estimates, we can perform heuristic clustering of the clonal, subclonal, and neutral tail mutations observed in tumour VAF distributions. (Methods). Heuristic clustering uses the theoretical observation that tumour VAF distributions, at minimum, must be composed of a neutral tail, if sufficient sequencing depth is achieved, and a clonal peak, representing mutations acquired prior to tumour initiation or fixed during cancer evolution. Therefore, by first assigning the frequencies of any detected subclones, we can then cluster subclones, the neutral tail, and clonal regions outward from these frequencies using the variance under any preferred sequencing noise model (e.g., binomial variance). We note here that this heuristic approach does not require any additional explicit clustering methods that require model selection and simply uses the expected variance under a given probability distribution. Notably, this heuristic approach facilitates subclonal clustering at almost zero additional computational cost (as observed in the total runtime per sample of ~1s).

We next evaluated TumE in whole-exome sequenced (WES) mismatch repair deficient (MMR) gastro-esophageal tumours biopsied across multiple spatially distinct regions (collected from von Loga et al. [33]). As evolutionary inference requires high-quality genomes, we only included samples that had a mean effective coverage (mean sequencing depth * purity) greater than 60x and a minimum purity of 50%. We note that ~70x mean sequencing depth has been suggested as the minimal threshold for accurate estimates [7,16], as we also observed (S8 and S9 Figs). Following removal of low-quality biopsies, we retained biopsies from two tumours with one tumour retaining 5 spatially distinct (WES) biopsies. TumE estimates in the 5 spatially distinct biopsies from a single tumour revealed the fixation process of a positively

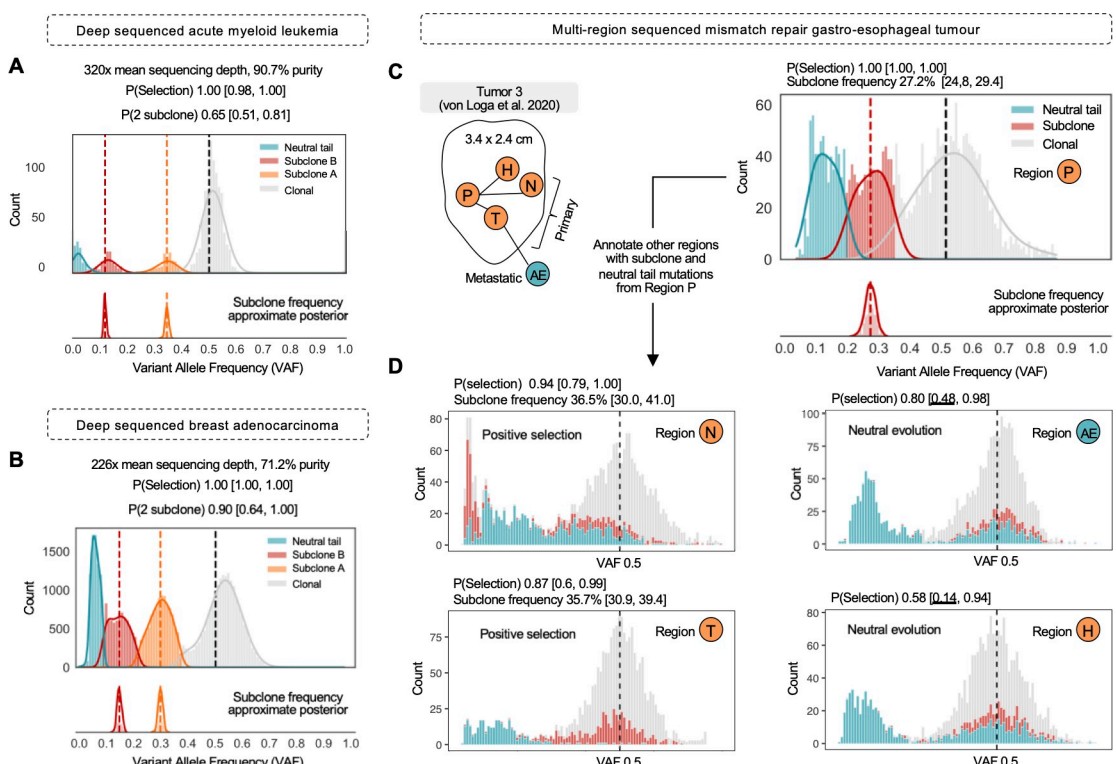

**Fig 3. TumE estimates in deep whole-genome or whole exome sequenced tumour biopsies. (A)** A deep-sequenced primary acute myeloid leukemia (AML) sample from Griffith et al. [31]. TumE estimated two subclones, a neutral tail, and a clonal peak. P (Selection) indicates the probability of selection. P(0, 1, 2 subclone) indicates the probability estimate for the number of subclones. Each probability estimate is provided with the 89% equal-tailed interval generated from 50 Monte Carlo dropout samples. A sample is labeled positive selection if the lower bound of the 89% interval is above P = 0.5, and the number of subclones is assigned to a sample if the lower bound of the 89% interval is greater than 0.5 (Methods). Subclone frequency estimates are shown with the complete approximate posterior. **(B)** A deep-sequenced breast adenocarcinoma from the pan-cancer analysis of whole genomes (PCAWG) [11]. TumE estimated two subclones, a neutral tail, and a clonal peak. **(C)** We applied TumE to a single mismatch repair deficient (MMR) gastro-esophageal tumour [33] sequenced across 5 spatially distinct regions. We first identified an intermediate frequency subclone in region P with TumE. **(D)** Under the hypothesis that TumE could reveal the fixation process of region P subclones in other regions, we annotated each of the remaining regions with the clonal, subclonal, and neutral tail mutations identified in region P. We identified ongoing subclonal selection in 2 out of the 4 remaining regions (N and T) consistent with an increase in frequency of subclonal and neutral tail mutations from region P. In cases where neutral evolution was the most parsimonious explanation, we observed complete fixation of the region P subclonal mutations (region AE and H).

selected subclone, from intermediate frequency to metastasis fixation (Fig 3C and 3D). In addition, the application of TumE to multi-region samples highlighted the ability of TumE to pick up signatures of selection not directly encoded in distinct subclonal peaks but in the asymmetry of the diploid heterozygous cluster (region N & region T, Fig 3D).

Finally, we evaluated TumE in 78 whole-genome sequenced (WGS) tumour biopsies with >60x mean effective sequencing depth, spanning 10 different cancer types, retrieved from the pan-cancer analysis of whole genomes (PCAWG) [11]. To ensure valid inference with TumE, we only used SNVs from high-quality consensus diploid regions and performed purity correction and adjustment (Methods). In total, TumE identified evidence for positive, or subclonal, selection in 50% of samples whereas the other 50% were adequately described by neutral evolutionary dynamics (Fig 4A and S2 Table). Analysis of diploid point mutations in samples with at least one detected subclone (n = 39) revealed that 26% of samples carried a nonsynonymous, indel, or splicing mutation in a known driver gene. The observation that only a minor fraction of detected subclones harbored point mutations in known drivers is not unexpected as

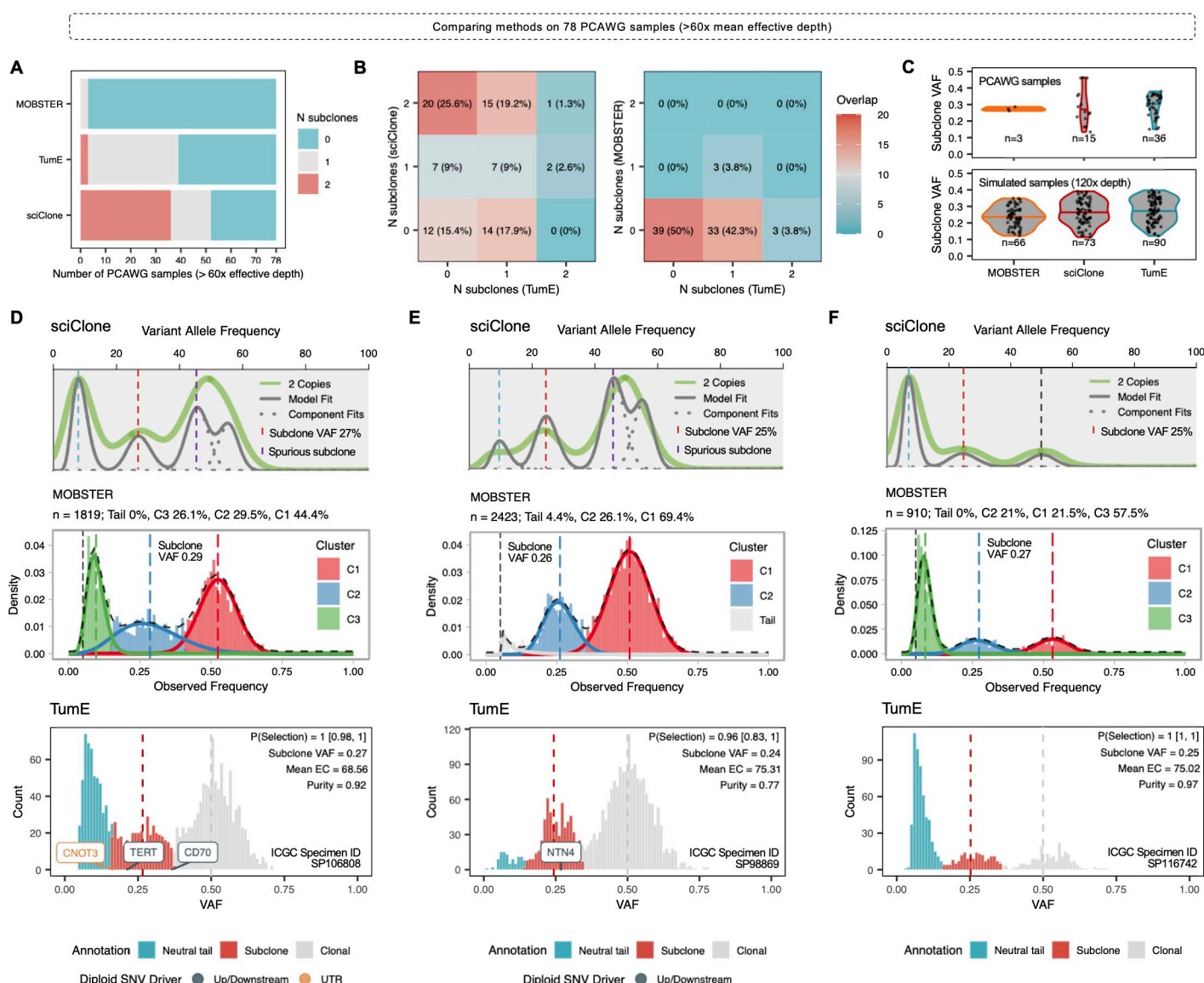

**Fig 4. Comparison of subclone detection methods in 78 PCAWG samples with mean effective sequencing depth (purity * depth) of greater than 60x. (A)** Number of subclone estimates using different methods including MOBSTER [16], sciClone [24], and TumE. **(B)** Comparing agreement and disagreement between existing methods in the analyzed PCAWG samples. **(C)** Comparison of the estimated subclone frequency in PCAWG samples versus simulated data across different methods. Consistent with simulated data where each method properly identified the presence of 1 subclone, TumE consistently captures higher frequency subclones when compared to MOBSTER. **(D–F)** Visualization of the 3 samples where each TumE and MOBSTER detected 1 subclone. sciClone fits are provided for comparison. Single-nucleotide variants (SNV) with annotations in known driver genes [8,16] were labeled if present.

previous work on timing driver mutations has suggested the majority of driver point mutations occur early in tumor development [34]. In this regard, the elevation in subclone frequency may also be driven by a plethora of alternative genomic alterations, such as structural variation, not analyzed here. VAF distributions annotated with TumE fits and driver mutations can be found in S19 Fig.

Next, to facilitate direct comparison against existing methods, we also applied MOBSTER and sciClone to the 78 WGS samples. Consistent with analyses in simulated tumours, MOBSTER tended to more parsimonious explanations of the data (4% selection, S3 Table) while

sciClone (corrected for presence of a neutral tail) estimated excess selection (67% selection, S4 Table) relative to TumE (Fig 4A and 4B). Overall, the general agreement between accurate classification in simulated tumours and estimates in empirical tumours suggested TumE was likely detecting true cases of ongoing selection. When compared to MOBSTER in both simulated and empirical data, the additional subclones detected by TumE can likely be explained by the ability to detect a wider range of subclone frequencies, particularly higher frequency, (Fig 4C) that evade detection by mixture models with flexible beta distributions and model selection strategies that penalize overlapping clusters [16] (Fig 4C). The identification of additional higher and lower frequency subclones is also consistent with previous theoretical work showing that subclones have a higher probability of being at more extreme or 'peripheral' frequencies in growing tumours [14].

Previously, alternative methods applied to large cancer cohorts, including PCAWG, have estimated that as few as 3% [16] to upwards of 95% [35] of samples show evidence for ongoing subclonal selection. The discrepancy is likely explained in modeling approaches and sample selection strategies. For example, in terms of modeling approaches, low estimates are a by-product of utilizing mixture models and model selection strategies that rely on non-overlapping and 'clean' subclonal peaks whereas high estimates likely occur from not considering the neutral dynamics in tumour evolution. In contrast, TumE generates a non-linear encoding of the VAF distribution, extracting novel representations that increase accuracy while simultaneously accounting for the correct neutrality evolutionary dynamics observed in tumour populations. By predicting subclone frequencies first, clustering is a natural post-hoc benefit of the synthetic supervised learning approach.

In addition, the discrepancy in percentage of samples subject to positive selection across recent studies may also be a by-product of cohort selection and sample filtering. We note that in this study we only included PCAWG samples with >60x mean effective sequencing depth as it represented thresholds for accurate inference of selection by TumE. This differs from previous work by Caravagna et al. 2020 (MOBSTER) [16] where samples were selected based on a post-clustering metric, namely the presence of a neutral tail that carried a relatively large proportion of mutations (>10%). However, because the majority of the PCAWG samples with >10% neutral tail are not sequenced deep enough (effective depth < 60–70x) to mitigate overdispersion, detecting selection in these samples is either prevented or confounded. We believe selecting samples based on data quality required for detecting selection (e.g., effective depth), rather than post-hoc filtering based on fits, is likely the better strategy for properly quantifying the occurrence of ongoing selection. This disagreement in rates of positive selection was also highlighted in this study as only 1 sample was consistently called subclonal across TumE, MOBSTER and sciClone. We present the 3 overlapping samples between MOBSTER and TumE in Fig 4D–4F. Overall, our extensive analyses in simulated and empirical tumours suggests that methods considering the neutral evolutionary dynamics, such as TumE and MOBSTER, should be preferred over consistently incorrect methods such as sciClone. However, mixture models using beta distributions and conservative model selection strategies that penalize overlapping clusters (such as MOBSTER) may not be sufficient to capture the full range of subclone frequencies and may represent the baseline rate of ongoing selection in growing tumours.

## A transfer learning framework to infer additional evolutionary parameters

One drawback of simulation-based deep learning approaches is the requirement for the repeated generation of synthetic data for training. Although this allows for fast inference at test time through amortization, altering the model's assumptions or changing the parameters being inferred generally requires simulating a completely new set of data and training an

entirely new set of models—a computationally expensive process. Practically, overcoming this limitation would provide substantial reductions in the amount of time and data needed to build accurate models and would make simulation-based approaches more accessible to the general user. Therefore, we hypothesized that our trained deep learning models could be used as a source of 'stored' knowledge for related evolutionary inference tasks that also used the VAF distribution as input.

To explore this possibility, we implemented a transfer learning pipeline, based on domain adaptation [36,37], to make inferences on additional parameters using a previously developed cancer evolution simulator, TEMULATOR [38], that was built under a similar but modified set of assumptions relative to our multiplicative fitness framework (Methods, viable parameter combinations for detectable subclones outlined in S20 Fig). In this study, we employ open set domain adaptation [37] where the structure of the input space, i.e., the VAF distribution, is retained whereas the outputs, the evolutionary tasks, are modified. Briefly, this pipeline involved generating new synthetic tumour sequencing data using TEMULATOR, performing architecture 'renovation' on pre-trained TumE neural networks to replace existing task-specific branches with new ones, and re-tuning the neural network weights and hyperparameters for optimization on the new evolutionary inference tasks (Fig 5A). The evolutionary inference tasks included predicting subclone fitness, subclone emergence time, mutation rate, and subclone cellular fraction (subclone VAF * 2). To highlight the benefit of using pre-trained models on related evolutionary inference tasks, we opted to update network weights with only 500,000 synthetic VAF distributions, representing only a fraction (~1.25%) of the data used in the original training of TumE. Each VAF distribution was generated by simulating synthetic tumours with TEMULATOR at a birth rate of 1, death rate of 0.2, final population size of ~$10^4$, and with either 0 or 1 detectable subclone. The remaining parameters, such as mutation rate, were uniformly sampled from empirically plausible ranges (S5 Table).

Initially, we used the 500,000 synthetic VAF distributions to compare pre-trained vs non-pretrained models for predicting the evolutionary and subclonal parameters in the presence of 1 subclone. To ensure valid comparisons, we performed a random hyperparameter search, tuning the learning rate and number of fully connected layers in the new task specific branches across both the pre-trained and non-pretrained model groups. Both groups shared identical neural network architectures. When initially evaluating ~300 pre-trained and non-pretrained models on an external test set of 3000 synthetic tumours, we found that pretrained models obtained significantly lower loss (mean absolute error across all tasks, two-sided Wilcoxon test, $p < 2.22 \times 10^{-16}$, Fig 5B). Further, when evaluating the top performing pre-trained and non-pretrained models on an additional 400,000 synthetic tumours, pre-trained models obtained significantly lower mean percentage errors, relative to non-pretrained models, for predicting the mutation rate, subclone emergence time, subclone fitness, and subclone frequency (two-sided Wilcoxon test, $p < 1.7 \times 10^{-8}$ on all tasks, S21 Fig). We note that although 500,000 samples appeared to reach asymptotic predictive performance (S22 Fig), we have found that as few as 25,000 synthetic samples can generate a model with comparable predictive performance assuming the hyperparameter search space is large enough (e.g., >100 models; S22 Fig).

Next, we selected the top performing pretrained model, TumE transfer (TumE-T), for further validation. We initially found a modest yet systematic underestimation of the mutation rate (~50% mean percentage error). However, this was easily corrected with a post-hoc adjustment by re-fitting the predicted mutation rate to a set of 1000 synthetic tumours using polynomial regression (degree = 2). Evaluating the updated mutation rate estimates on a holdout set of 100,000 synthetic tumours validated the post-hoc adjustment (S23 Fig). Overall, we were able to effectively recover all evolutionary parameters in the 100,000 synthetic tumours with

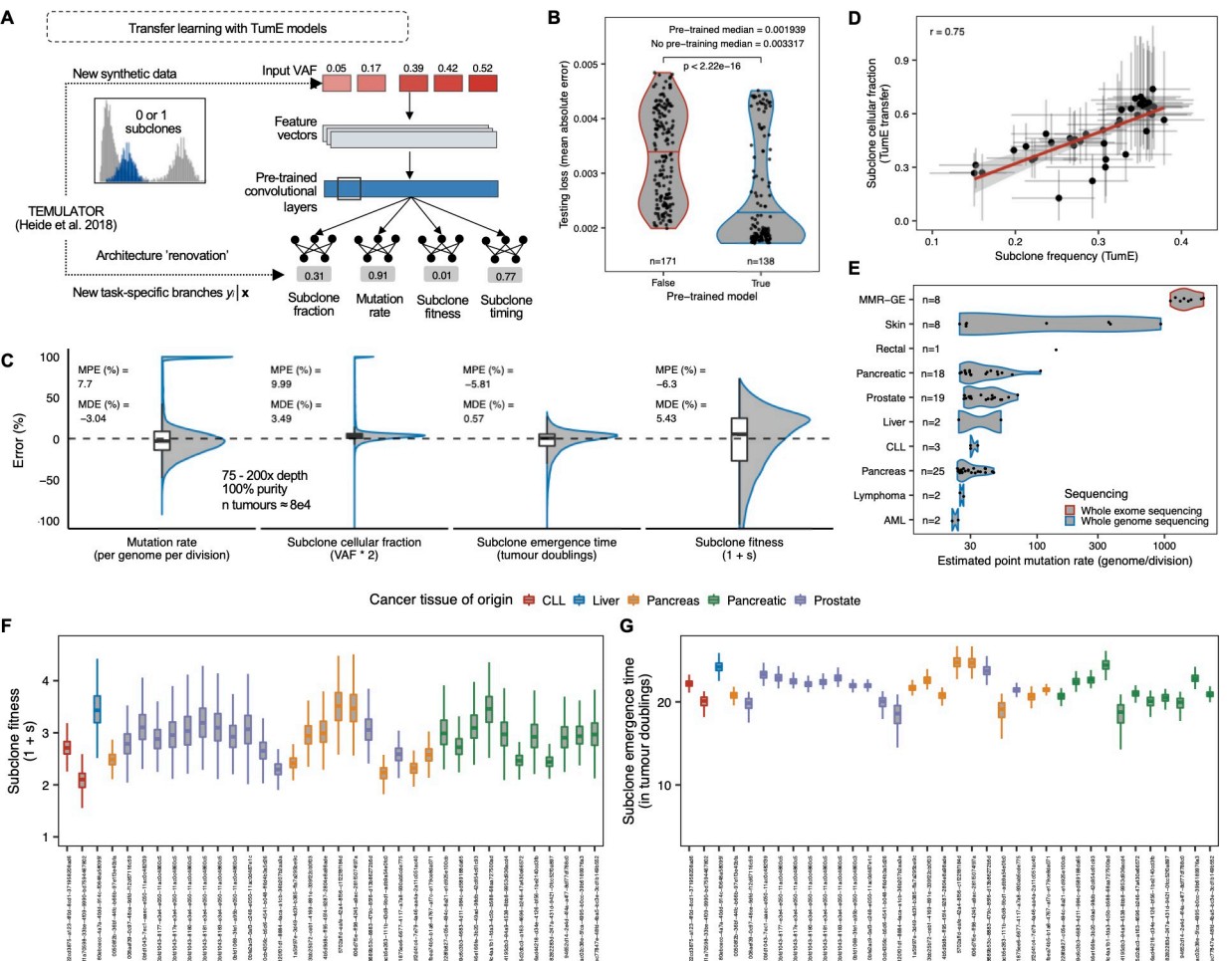

**Fig 5.** **(A)** Transfer learning approach utilizing 'renovated' pre-trained neural networks for alternative evolutionary inference tasks in tumour cellular populations. TEMULATOR is an alternative cancer evolution simulator that generates synthetic tumour sequencing data by deterministically initiating subclones at user specified fitnesses and time points [38]. **(B)** Pre-trained models provide significant reductions in testing loss, over non-pretrained models, when updating neural network weights on reduced dataset size of 500,000 synthetic VAF distributions (~1.25% of the total dataset size used to originally train TumE). **(C)** TumE transfer (TumE-T) effectively recovers evolutionary parameters from TEMULATOR simulations (75–200x mean sequencing depth, 100% tumor purity) with mean and median percentage errors less than 10% in all cases. A full description of performance across variable sequencing depths, mutation rates, and subclone frequencies is provided in S23 Fig. **(D)** We find consistency between the subclone cellular fraction estimated by TumE-T and the subclone frequency (cellular fraction / 2) estimates generated from TumE, indicating nearly identical tasks are easily transferred through pre-training. **(E)** Per genome per division mutation rate estimates in 88 WES and WGS samples from von Loga et al. [33] (MMR-GE = mismatch deficient repair gastro-esophageal cancer), Griffith et al. [31] (AML = acute myeloid leukemia), and PCAWG [11]. **(F)** Subclone fitness (1+s) estimates (relative growth rate advantage of subclone over background population) and **(G)** subclone emergence time estimates in 36 tumour biopsies identified with 1 subclone in the PCAWG data. Subclone fitness and emergence time estimates were scaled to a final tumour population size of $10^{10}$ cells, similar to [7]. PCAWG sample identifiers are provided on the x-axis.

mean and median percentage errors lower than 10% in all cases (Fig 5C). The performance was also consistent across sequencing depths and mutation rates, however, as expected, we could only effectively assign subclonal parameters, such as fitness, at detectable subclone frequencies (~10–40% VAF; S24 Fig).

Applying TumE-T to the 88 WGS and WES samples described above, we found strong correlation between predicted subclone cellular fraction and subclone frequency estimated by the original TumE models, suggesting nearly identical tasks are easily transferred to new source-target distributions when using pre-trained models (Fig 5D). With respect to mutation rates,

estimates were consistent with the general trends observed empirically [11,33]—with mismatch repair deficient tumours showing extremely high mutation rates (>1000 per genome per division) and acute myeloid leukemia showing very few (Fig 5E). For subclone fitness and subclone emergence time estimation, we had to consider the difference between simulated and true population sizes [7,19]. In this regard, we rescaled our estimates to account for a true tumour population size of $10^{10}$, similar to [7]. With rescaling, TumE-T subclone fitness estimates, defined as the relative growth advantage of the selected subpopulation over the background population, ranged from ~2.1 to 3.5 (Fig 5F) while subclone emergence time estimates ranged ~18 to 25 tumour doublings (Fig 5G) in samples with ongoing subclonal selection. We note that emergence times of ~18 to 25 tumour doublings represent approximately 0.001 to 0.16% of the final tumour volume, which is consistent with theory and empirical evidence suggesting subclones must arise early during tumour growth to reach detectable frequencies [7,14,17,39].

## Discussion

In this study, we developed a synthetic supervised learning approach, TumE, for cancer evolutionary inference. Overall, the synthetic supervised learning approach, TumE, provides four major advantages. First, by generating synthetic data using models of cancer evolution, we are able to explicitly account for the neutral and non-neutral evolutionary dynamics observed in tumour VAF distributions [7,12,16], thereby avoiding systematic overestimates in the number of subclones due to misclassifying low frequency neutral 'tails'. Second, by using neural networks that can naturally handle high-dimensional VAF distributions as input, we avoid information loss that comes with compressing data into a single statistic, or distance metric, prior to inference, improving model accuracy across all evolutionary inference tasks considered here. Third, by separating simulation and model training from prediction, via amortized inference, we significantly decrease inference time per sample, reducing time from minutes to seconds relative to existing methods. Finally, we show how we can use open set domain adaptation [36,37], a form of transfer learning, to recycle our models for alternative evolutionary inference tasks that use VAF distributions as input—drastically reducing the number of synthetic samples and computational time required for further model development. Lastly, in contrast to phylogenetic approaches that use mutations to reconstruct or explain the possible history of divergent clonal events in a tumour biopsy, population-genetic based methods, such as TumE, discriminate detectable subpopulations subject based on explicit evolutionary models such as positive selection or neutral evolution using the VAF distribution [40,41]. Altogether, our library of pre-trained models benefits all researchers building inference machines for cancer evolution prediction, even in a non-deep learning setting. For example, providing fast, evolutionary-informed peak initializations for mixture model-based methods.

We mention some current limitations. Firstly, as a neural network requires optimization on a finite, static set of data, estimates, without transfer learning, are constrained to a pre-defined search space. In this study we focused on cancer evolution in the context of 2 detectable selected subpopulations captured from frequency information in diploid genomic regions. Although multiple studies have shown it's rare to detect 2, or even 1, subclones [7,14,16,17] in noisy one-dimensional VAF distributions, it's possible we do not capture extreme cases of selected subclonal heterogeneity. Furthermore, focusing on diploid regions may obscure the detection of ongoing selection if mutations are concentrated in copy number aberrated segments. However, constraining analyses to diploid regions provides a strong baseline for model development, while genome-wide linkage provides biological justification for analyzing diploid regions. We also note that copy number variants miscalled as diploid can lead to overestimation of

subclonality by introducing spurious mutations in the subclonal frequency range. However, in this study, we focused on using high-quality consensus copy number calls extensively validated in pan-cancer cohorts and utilized in previous work used to parse out the evolutionary history of tumours [11,16,34]. Finally, our model of tumour evolution was structured to reflect the biopsy material available here, namely bulk sequenced single site and time point data. We note that tumour growth over space and time can have profound effects on the detectability of selection [42,43]. In this regard, TumE estimates can still be applied in a localized setting and aggregated globally. Nevertheless, more structured ways of integrating a synthetic supervised learning approach with multi-region data are necessary for maximizing utility.

Altogether, in this study, we exhibit how coupling well-specified synthetic data with neural networks provides fast and accurate amortized estimates that go beyond the current paradigm of single statistics, mixture models, and approximate Bayesian computation for classifying and quantifying ongoing selection in tumour populations. The integration of generative and simulation-based models of cancer evolution with modern deep learning frameworks facilitates robust and efficient estimates of evolutionary and subclonal dynamics in growing tumour populations. This extensible framework provides future avenues for harnessing progressive computational gain for the benefit of cancer genomics and, as an end goal, the cancer patient.

## Methods

### Synthetic data generation

We generated synthetic data that encoded the evolutionary dynamics observed in the variant allele frequency (VAF) distribution (namely the neutral tail, subclones, and clonal peaks) using two complementary approaches dependent on the underlying evolutionary mode—one for tumors subject to positive selection and one for tumours evolving neutrally.

For tumours simulated under positive selection, we utilized a well-established framework of cancer evolution that models exponential tumour growth under a stochastic branching process [7,12,13,15,19,29] and coupled this with a virtual biopsy procedure to account for sequencing noise observed in whole-genome/whole-exome sequenced tumours from real patient tumours. For implementation, we adapted a previous cancer evolution framework developed by Williams et al. [7]. Briefly, this model simulates exponentially growing tumour populations under a stochastic branching process using a rejection-kinetic Monte Carlo (MC) algorithm, where a given cell accrues mutations at some Poisson-distributed per genome per division rate $\mu$ and divides or dies with probabilities proportional to its birth or death rate. This branching process continues by randomly sampling existing cells, weighted by cellular fitness, until a final tumour population size $N$, sufficient to recapitulate the features of empirical VAF distributions, is reached. Following completion of each simulation, a virtual biopsy procedure to account for sequencing noise observed in real patient VAF distributions is implemented. In this sequencing noise model, the observed frequency for a given mutation ($VAF_{obs}$) relative to the true underlying frequency ($VAF_{true}$) in a tumour of population size $N$ is given by

$$VAF_{obs} = R_{obs} \, / \, D_{obs}$$

where

$$D_{obs} \sim Bin(n = N, \; p = \frac{D}{N}), \; R_{obs} \sim BetaBin(n = D_{obs}, \; p = VAF_{true}, \; \rho)$$

where $D$ total indicates the total observed read depth, $R$ indicates the number of observed reads covering the mutation, $VAF_{true}$ indicates the true population frequency of the mutation, and $\rho$ indicates the overdispersion parameter for the beta-binomial.

In this study, we modify the Williams et al. [7] framework in two ways. Firstly, we implement a fully stochastic arrival of subclones (driver mutations) rather than deterministically injecting a subclone with a specified fitness at a given time $t$. Secondly, the fitness of a subclone or cell is dictated by the multiplicative fitness of all driver mutations. Therefore, when a driver mutation does occur, based on some probability $p_d$, it is assigned a selection coefficient $s > 0$ sampled from an exponential distribution which increases the cell's growth rate ($b$–$d$) by a factor of $(1 + s)$ i.e., the fitness. In the case of multiple driver mutations, the fitness of a given cell increases multiplicatively i.e, $\Pi(1 + s)$. Although this random injection of driver mutations is more computationally intensive, it implicitly captures a wider variety of potential frequency distributions without hard coding additional parameter settings—for example, when additional subclones, beyond 1 or 2, are present at undetectable frequencies (e.g., >40% or <10%). In this study, we consider up to 2 detectable subclones but allow for up to 3 selected subclones to be present at the time of biopsy (see Simulation Parameter Selection below for more details).

For tumours simulated under neutral evolution, we use a generative sampling process for producing neutral VAF distributions, rather than using the stochastic simulation framework. We implement this sampling process because we use a small $N$ population size approximation to generate VAF distributions in our stochastic simulations (using a small $N$ allows us to increase simulation speed and efficiency, which makes generating millions of synthetic VAF distributions practically feasible). Although using a small $N$ is reasonable since the VAF distribution contains no information on population size [7] (a final simulated tumour population size $N$ of $10^3$–$10^4$ has been shown to be sufficient to recapitulate the properties of empirical VAF distributions [7]), neutral stochastic simulations have a higher probability of returning late-occurring spurious subclones due to chance or, in empirical terms, genetic drift. Given the quality of the synthetic data impacts deep learning model performance, we utilize the fully synthetic generative sampling scheme to avoid misspecified data relative to the expected null model of neutral evolution.

The neutral generative sampling process we implement is based on the observation that neutrally evolving asexual, non-recombining populations, such as cancer populations, have VAF distributions (excluding clonal mutations) that follow a power-law or Pareto distribution [16,30]. Therefore, a VAF for any mutation $i$ in the neutral tail of a frequency distribution can be realized by sampling

$$VAF_i \sim f(x \mid \alpha,\ m) \text{ with} f(x \mid \alpha,\ m) = \alpha m^\alpha x^{-(\alpha+1)}$$

where ɑ is the shape parameter and $m$ the scale parameter for the Pareto distribution.

Given that the generative process for neutral tails is known [16,30], if we have empirically valid shape and scale parameters that define the Pareto distribution, we can generate realizations of neutral allele frequency distributions that are well-specified. Previously, Caravagna et al. [16] fit Pareto distributions (and beta distributions) to thousands of patient tumours extracting both shape and scale parameters. We used these Pareto distribution fits from diploid regions of patient tumours with greater than 50x sequencing coverage to build sampling distributions for the shape and scale parameters. We then used these sampling distributions to generate allele frequencies under a Pareto distribution and, in addition, randomly assigned clonal mutations to each neutral synthetic VAF. In practice, as previously noted [16], the scale parameter can be set to the minimum observed frequency as this is the maximum likelihood estimate for the Pareto distribution.

We note that we added additional noise to synthetic neutral distributions to better account for variability observed in empirical data in two ways. Firstly, for any synthetically generated neutral distribution, we randomly trimmed the low frequency neutral tail at a frequency $f$ (10–

30% VAF) with some probability $P_{trim}$ ($\leq 0.1$). We perform this step as many VAF distributions observed in patient biopsies lack the characteristic neutral tail, even at high sequencing depth [16]. By randomly trimming neutral synthetic VAF distributions, we tend to more parsimonious explanations of the data, with respect to positive selection, when assessing incomplete and potentially noisy VAF distributions. Furthermore, by adding sequencing overdispersion following trimming, we naturally capture the observed downsampling of degenerate neutral tails in empirical data. Secondly, we randomly shifted the heterozygous, diploid clonal peak (that should be centered at 50% VAF) to between 45 and 50% VAF. We perform this random perturbation of the clonal peak to avoid overestimating positive selection when patient samples have incorrect tumour purity estimates that may have led to spurious elevation in the number of subclonal mutations.

Finally, to ensure positively selected and neutrally evolving tumours were not out of distribution with each other given the alternate data generation approaches, we built an aggregate simulation framework that generated neutral and positive synthetic tumours in pairs—assigning the neutral VAF distributions with parameter-matched sequencing noise and equivalent clonal and non-clonal mutations with respect to the paired positive selection simulation.

The synthetic data generation algorithms are outlined in S1 Text and code is available at https://github.com/tomouellette/CanEvolve.jl.

## Simulation parameter selection

Each stochastic simulation described above was parametrized by the mutation rate (per genome per division), the probability a mutation was a driver, the mean for the exponential selection coefficient distribution, the number of clonal mutations in the founder cell, the maximum number of driver mutation events, the final tumour population size, the sequencing depth, and the sequencing overdispersion. We chose simulation parameters that were consistent with previous studies [7,13,16] and that captured the expected qualitative and quantitative attributes of VAF distributions observed empirically (S1 Table). All non-fixed parameters were uniformly random sampled during the development of the synthetic datasets. To improve computational speed and efficiency in our stochastic simulations, we used similar simulation approximations as [7]. Namely, a small $N$ population size approximation (where we simulated data using a final population size of $10^3$) and a fixed growth rate (where the birth rate was set to log (2) and the death rate was set to 0). In addition, as we were focused on differentiating between neutral evolution and selection at effective sequencing depths of ~50–250x, we constrained our search space to 1 or 2 detectable subclones present between 10–40% VAF. We implemented this constraint as (i) it is extremely rare to detect 3 subclones in a one-dimensional VAF distribution as each subclone has to be >5–10% VAF (10–20% cellular fraction) for detection, (ii) most frequency-based methods or studies show limited evidence for detecting >1 subclone at 50–250x coverage in a single time point, one-dimensional VAF distribution [16], and (iii) below greater than roughly 10% VAF subclones merge with the neutral tail and above roughly 40% VAF subclones begin to merge with the clonal peak when considering diploid regions.

## Synthetic supervised learning

As outlined in the results, synthetic or simulation-based deep learning has been shown to be equivalent to amortized approximate inference under a generative model [28]. Therefore, by optimizing a neural network using synthetic VAF distributions sampled from a stochastic generative process $p(\mathbf{x},\mathbf{z}|\boldsymbol{\theta})$ (i.e. the synthetic data generation scheme defined above), we can build

fast approximate inference models for evolutionary inference. We describe the synthetic supervised workflow from feature generation to prediction below.

### Input representation

For each simulation, we converted mutation frequencies into a VAF distribution (histogram) of length $k$ that had a fixed range between 2% and 50% VAF. To implicitly condition our model on mean sequencing depth (readily available from sequenced tumour biopsies), we only included mutations above a frequency cutoff based on the variance of a binomial sequencing noise model. We note that this conditioning step is arbitrary and simply acts to improve model optimization during training. In general, a simple approach to conditioning a neural network on a measurable variable involves finding a reasonable encoding within the feature representation. For example, an alternative approach instead of using a frequency cutoff would be to concatenate the sequencing depth to the input feature vector. Overall, each input feature vector was created by counting mutations into $k$ bins where each bin had a width $w$ of (50–2% VAF) / $k$ and a lower frequency cutoff defined by $f_{alt} + \left(2\sqrt{f_{alt}c[1 - f_{alt}]}\right) / c$ where $f_{alt}$ is the minimum alternative reads to call a mutation divided by mean sequencing depth and $c$ is mean sequencing depth. For all model development and training, we generated and concatenated two feature vectors with $k = 64$ and $k = 128$ for each simulation to capture varying levels of information depending on the sparsity of mutations in each synthetic tumour.

### Model search

We initially developed neural networks for three single or multi-task inference problems: (i) evolutionary mode (neutral evolution or positive selection) and number of subclones classification ($M_{ms}$), (ii) frequency prediction for a single subclone ($M_{1s}$), and (iii) frequency predictions for two subclones ($M_{2s}$). For each multi-task, we performed a random hyperparameter search using a one dimensional (1D) convolutional neural network (CNN) with task-specific fully connected branches as a base architecture. For the random search, the hyperparameters included the number of convolutional layers (1–20) the task-specific branch type (fully connected or global average pooling), the number of feature maps/channels for each convolutional layer (4–32), the convolutional kernel width for the left trunk, right trunk, and task-specific branches (1–17, odd), the learning rate ($10^{-7}$–$10^{-3}$), and the patience for early stopping (3–5). To tend toward higher precision and lower recall for predicting selection, we also tuned a penalty term on the positive class in the binary cross entropy loss. Batch size was fixed to 256. Hardswish activations were used at each hidden layer. Dropout, fixed at a probability of 0.5, was added after each layer to allow for downstream application of uncertainty estimation (see Uncertainty Estimation below). We note that we also explored inferring subclone emergence time under a multiplicative fitness model but could not effectively recover parameters likely due to a complex non-linear relationship between subclonal fitness and emergence time. However, we provide these estimates as an 'experimental' output in the TumE python package (links below).

### Model training

We trained over 150 models for each evolutionary inference task(s) using an Adam optimizer, minimizing the cross-entropy loss for classification tasks ($M_{ms}$) and the L1 loss for regression tasks ($M_{1s}$ and $M_{2s}$), on approximately 40 million synthetic tumours simulated with parameters outlined in S1 Table. For training, each batch consisted of 20,000 unique simulations and training was stopped after 4 epochs or when early stopping, updated after each batch, was

activated based on specified patience. To avoid overrepresentation of any subclone frequency during training, we re-balanced positive selection simulations before each batch to have an equal number of subclones at each frequency up to two decimal places (e.g., 0.11 or 11% VAF). For two subclone simulations, we re-balanced simulations based on the distance between subclones ($|f_{subclone2} - f_{subclone1}|$) and only included simulations where the distance between subclones was >5% VAF. Note that only positive selection simulations were used to train $M_{1s}$ and $M_{2s}$.

## Model selection

Using an independent test set of one hundred thousand simulations, we then selected the top models across each multi-task for further validation. For $M_{ms}$, we selected models that maximized the mean accuracy across the evolutionary mode, *P(Selection)*, and number of subclones, *P(N subclones)*, classification, and favoured models that assigned a larger penalty term to misprediction of positive selection (i.e. a lower weight to the positive class in the binary cross entropy loss). For the regression models $M_{1s}$ and $M_{2s}$, we selected models that minimized the mean absolute error between the true and predicted subclone frequency on the test set while also ensuring that predictions properly extrapolated across the entire simulated parameter range (e.g., ~10–40% VAF for subclone frequencies).

## Uncertainty estimation

To capture model-based uncertainty in our estimates, we implemented a form of Bayesian approximation for deep learning called Monte Carlo (MC) dropout [26,27]. Conceptually, MC dropout captures model-based uncertainty by taking advantage of the relationship between model averaging and standard dropout—a network with dropout at every layer encodes $2^n$ possible network configurations. By keeping dropout on at test time, each prediction is a stochastic pass through a set of randomly activated neurons. More specifically and with a slight abuse of notation w.r.t to ref [26] ignoring the variational notation, we make estimates of our target variable *y* (e.g. subclone frequency) by performing *T* stochastic forward passes through the network and averaging, *E(y)*, the results:

$$E(y) \approx \frac{1}{T} \sum_{t=1}^{T} \hat{y}(x, W_1^t, \ldots, W_L^t)$$

where $\hat{y}$ is the output with respect to the input data *x* for a neural network with *L* layers, and *W* corresponds to a weight matrix for each layer *L*. For every stochastic pass, each *W* is assigned a randomly sampled vector of Bernoulli random variables such that each individual neuron is inactivated with a probability equal to the dropout rate. Under this framework, MC sampling over *T* stochastic passes through the network generates an approximate posterior for our target variables with respect to the input data.

## Making predictions

For differentiating between neutral evolution and positive selection, P(*Selection*), and predicting the number of subclones, P(*N subclones*), in both synthetic and real patient tumours, we took a conservative, more parsimonious approach to prediction by considering the variance in the approximate posterior. For P(*Selection*), we only called positive selection if the lower bound of an 89% equal-tailed interval for the approximate posterior, computed across 50 stochastic passes through $M_{ms}$, was greater than 0.5. If the lower bound was less than 0.5, we called neutrality and zero subclones, independent of the result of P(*N subclones*). We show the

utility of this strategy for mitigating model overconfidence in a synthetic toy example (S6 Fig). For the regression tasks of predicting subclone frequency and emergence time, we estimated the true value by performing 50 stochastic passes through the networks and averaging the results, while also providing the complete approximate posterior. We describe additional considerations for making estimates in real patient tumour biopsies below.

All model development and training were done using *pytorch v1.8.1*. We provide a python package, scripts, and all trained neural network models for downloading, use, and modification at https://github.com/tomouellette/TumE.

## Model performance in synthetic tumour sequencing datasets

We simulated or collected 3 different datasets of synthetic tumour sequencing data to study the performance of TumE under changing parameter regimes or changes to model assumptions. The first dataset, generated by our simulation framework described above, consisted of ~2.8 million synthetic tumours simulated across varying sequencing depths and overdispersions (all parameters provided in S1 Table). Using this dataset, we compared TumE against six frequency-based summary statistics for differentiating between positive selection and neutral evolution. Four of the statistics were cancer evolution statistics developed previously [12] and provided in the R package *neutralitytestr*. For each sample, the parameters of *neutralitytestr* were set as follows: ploidy = 2, cellularity = 1, read_depth = simulated mean sequencing depth, rho = simulated overdispersion (rho). Two of the statistics were common population genetic statistics, Tajima's *D* [20] and Fay and Wu's *H* [21]. Only variant allele frequencies and sequencing depth were required for input to compute these statistics. We provide an implementation of Tajima's *D* and Fay and Wu's *H* for tumour sequencing data in the github repository. We additionally evaluated a mixture model-based approach MOBSTER [16] for subclone detection and frequency quantification. To enable analysis of ~2.8 million synthetic tumours, we ran MOBSTER with the following parameters: K = 1:3, samples = 2, init = "peaks", tail = c (TRUE, FALSE), epsilon = 1e-6, maxIter = 100, fit.type = "MM", seed = 12345, model.selection = "reICL", pi_cutoff = 0.02, N_cutoff = 10. We defined the number of subclones that MOBSTER detected as follows. If a tail and 3 beta components were fit then we assigned 2 subclones, if a tail and 2 beta components or if no tail and 3 beta components were fit then we assigned 1 subclone, and for all remaining fits we assigned 0 subclones or neutrality.

The second dataset was retrieved from Caravagna et al. [16] and consists of synthetic tumour sequencing data from 150 tumours sequenced to 120x depth using a beta-binomial sequencing model and grown to a size of $>10^8$ at a birth rate of 1 and death rate of 0.2. The complete description is provided in the supplementary of [16]. We used this dataset to evaluate the small N approximation and to compare TumE to existing mixture model methods. We applied both MOBSTER and a variational Bayesian mixture model sciClone [24] to this dataset. MOBSTER was run under default package settings without parallel computation and with K = 1 to 3 beta components. sciClone was run under default package settings with copyNumberCalls fixed to 2 and maximumClusters fixed to 4. To estimate the number of selected subclones with sciClone, which doesn't account for neutral evolutionary dynamics, we took the inferred number of subclones and subtracted 2 (representing the neutral tail and clonal peak). Per-sample runtimes for TumE, MOBSTER, and sciClone were computed on a single machine with 16GB memory and a 2.3GHz quad-core Intel i7 processor.

The third dataset was used to evaluate variable birth and death rates on TumE estimates for predicting positive selection, determining the number of subclones, and estimating subclone frequency. The dataset consisted of ~6 million synthetic tumours, generated by our simulation framework described above, grown at variable birth and death rate combinations. Mutation

rate and mean sequencing depth were both fixed to 100. Other parameters were uniformly sampled and all parameters evaluated are outlined in S1 Table.

The fourth dataset was used to evaluate the robustness of TumE estimates to variation in tumour purity. We simulated 6000 synthetic tumours at sequencing depths of 20–200x with a beta-binomial overdispersion of rho = 0.03 and tumour purities ranging from 20–100%. We then made an auxiliary dataset to assess the impact of incorrect purity estimates on TumE predictions. To generate this dataset, we added -25 to 25% error on each purity estimate. In total, 10 datasets with varying error on purity estimates were generated. We then performed purity correction and adjustment of the VAFs from each synthetic tumour prior to evaluating the false positive rate for positive selection with TumE. A more detailed outline of the purity correction and adjustment strategy is outlined below.

## Evolutionary parameter estimates in bulk sequenced single tumour biopsies

In this study, we used diploid regions of patient tumours for evolutionary inference as we did not have access to accurate phased mutation information for copy number correction of VAFs at non-diploid sites. However, in the absence of complete whole-genome duplication, mutated diploid regions should be sufficient to capture ongoing selection, due to selective sweeps from genome-wide linkage, if a sufficient number of neutral passengers mutations are accumulated during cell division over time [7,16]. We note that, similar to [16,34,35], we only used high-quality diploid consensus calls (star 3) from the original PCAWG study to avoid potential biases from miscalled segments appearing as subclones in the VAF distribution. Samples with fewer than 100 mutations following removal of low-quality regions were not included in this study.

In addition to only analyzing diploid regions, we only accepted tumour samples that had at least a 60x mean effective coverage (mean sequencing depth times purity). We set this cutoff as previous studies, and ours, have shown that tumour genomes sequenced below 50-70x coverage are exceedingly noisy and have insufficient limits of detection relative to low-frequency mutations for proper evolutionary inference [16,31].

Relative to our simulations, VAFs in bulk sequenced single tumour biopsies may also be confounded by impurity, where purity (cellularity) is defined as the percentage of cells in the biopsy that are of malignant or tumour origin. In general, low tumour purity can lead to spurious identification of subclones as it results in lower observed VAFs relative to the true underlying population VAFs. To ensure our inferences weren't biased by impurity, we corrected all VAFs using corresponding tumour purity estimates collected from the study of origin where $VAF_{corrected} = VAF_{observed}$ / purity.

We also note that some purity estimates may be incorrect—in these cases updating the VAFs with incorrect purity estimates can lead to a heterozygous clonal cluster (that should be centered at approximately 50% VAF) in the subclonal frequency range (~10–40% VAF). To ensure clonal clusters were properly centered at 50% VAF following purity correction, we performed additional adjustments to each patient's VAF distribution using the following heuristic. We first computed the density for each VAF distribution and then identified all the locations where the second derivative of the density was zero i.e., peak finding. If the closest peak to 50% VAF (the theoretical diploid clonal cluster) was above 35% VAF, we considered it a misrepresented clonal peak. We made this assumption as analyses in pan-cancer datasets suggest that all tumours are initiated in somatic cells already carrying mutations [10,11]. We then fit a Gaussian distribution to the identified clonal cluster of each patient VAF distribution and adjusted each VAF by multiplying by 0.5 divided by the mean of the fit. Although a Beta distribution is generally used for

fitting clonal clusters in cancer genomics, a Gaussian is a reasonable approximation for adjusting VAFs based on incorrect purity estimates as it provides accurate estimates of the cluster mean, and has been used in previous subclonal clustering methods [44]. The quality of this adjustment strategy (false positive rate for positive selection) was evaluated in 6000 synthetic tumours simulated across sequencing depths of 20–200x, tumour purities of 20–100%, and purity estimate errors ranging from -25 to 25% (S17 and S18 Figs).

Heuristic clustering using the estimated subclone frequencies was performed either using the expected variance under a binomial sequencing noise model or, alternatively, using the subclone frequency estimates to initialize the means of a gaussian mixture model. Clustering under the binomial framework was performed as follows. Given an estimated subclone frequency $q$, all mutations within the frequency range of $q \pm \left( \varepsilon \sqrt{qc[1-q]} \right) / c$ were assigned to the subclone, where $\varepsilon$ scales the cluster width and $c$ is the mean sequencing depth across the tumour genome or exome. We fixed $\varepsilon$ to 2 in this study.

For an empirical comparative analysis against existing methods, we fit both MOBSTER and sciClone to the 78 PCAWG samples. For MOBSTER, we used default settings that enabled fitting up to 3 beta components with a tail using an 'ICL' model selection strategy. We also attempted to use the 'reICL' model selection strategy; however, we were generating excessive positive selection calls relative to the original study and therefore opted for the ICL strategy. For sciClone, we used default settings as specified in the github repository (https://github.com/genome/sciclone). We report both ICL and reICL fits for MOBSTER in S3 Table and sciClone fits in S4 Table.

## Transfer learning for inference in alternative synthetic data regimes

Given a pre-trained network with weights optimized to a source domain $S$, composed of input space $X_S$, output space $Y_S$, transfer learning attempts to use pre-training to improve the performance on another target domain $T$ composed of $X_T$ and $Y_T$. We employ a variant of transfer learning called open set domain adaptation [37] to take advantage of our pre-trained models for additional inference tasks. In this case, the input spaces remain constant (Xs = Xt, VAF distribution) but the inferred tasks are allowed to differ. Open set indicates that some tasks may overlap with the output space of both the source and target domains.

To provide a concrete use case for transfer learning in synthetic supervised learning, we aimed to infer additional evolutionary parameters such as subclone fitness, subclone emergence time, mutation rate, and subclone cellular fraction (subclone frequency $*$ 2) using synthetic tumour sequencing data generated by an alternative cancer simulation framework TEMULATOR [38]. TEMULATOR differs from our synthetic data generation scheme, which was built around a multiplicative fitness driver model, as subclones are deterministically initiated at user specified emergence times and fitnesses. To facilitate transfer between previous and new tasks, we performed architecture renovation on the pre-trained neural networks, retaining all convolutional layers while replacing existing task-specific fully-connected branches with new task-specific fully connected branches (4 in total). To maximize the amount of information transferred to new tasks, we combined the convolutional layers from both the $M_{ms}$ and $M_{1s}$ models described above.

We then simulated 500,000 synthetic tumours at a birth rate of 1, death rate of 0.2, and final population size of $10^4$ (additional parameters such as mutation rate were uniformly sampled and are outlined in S3 Table). To facilitate efficient simulation, we first fit a noisy Gaussian process (GP) regression to the viable emergence time and fitness parameter combinations (that generated detectable subclones between ~10–40% VAF) and used the GP to sample viable emergence times given a uniformly sampled fitness. We made the GP noisy to facilitate

parameter combinations that resulted in subclones across the entire frequency range. The GP was fit using three kernels (RBF with length scale 100, dot-product, and white noise) and an alpha of $10^{-6}$ in the python package *scikit-learn v1.0*. Next, we used the simulations to re-optimize the pre-trained model weights, using an Adam optimizer to minimize the L1 loss (mean absolute error) for predicting new evolutionary inference tasks. To ensure fair benchmarking between networks with and without pre-trained weights, we performed a random hyperparameter search with ~150 pre-trained and ~150 non-pretrained models, tuning the learning rate and number of fully connected layers in the task-specific branches. Additional synthetic data used for evaluating performance was generated under similar parameter settings. We corrected modest yet systematic overestimates in mutation rate (~50% mean percentage error) in the final transferred model by fitting a polynomial (degree 2) ridge regression in *scikit-learn v1.0* to the predicted mutation rates. The mutation rate adjustment was performed using VAF distributions from 1000 synthetic tumours. We validated the correction on an additional 100,000 synthetic tumours. All TEMULATOR synthetic tumours were generated using parameter settings in S3 Table.

Predictions in empirical samples were performed by taking 500 Monte Carlo dropout samples and averaging the results. Dropout was only activated at test time on the new task-specific branches. Per-base mutation rate estimates in whole-exome sequenced MMR-GE samples were rescaled based on the 60MB Agilent SureSelectXT Human All Exon v6 kit used in the original study [33]. Because subclone fitness and emergence time is impacted by final tumour size, we rescaled our estimates to a realistic tumour size of $10^{10}$ cells, similar to [7]. Previous work [7] has shown that given subclone frequency $f_{sub}$ and an estimated final population size $N_{end}$, the age of a tumour at time of biopsy can be estimated by $t_{end} = log2([1—f_{sub}] * N_{end})$. Therefore, given that the relationship between emergence time in tumour doublings and log population size is linear, we can generate a rescale fitness estimate $w_R$ as follows.

$$w_R = 1 + (w - 1) * \frac{t_{end} - t_s}{t_{end_R} - t_{s_R}}$$

where $w$ equals subclone fitness, $t_{end}$ indicates time at tumour biopsy or final population size in tumour doublings, and $t_s$ indicates the time of subclone emergence in tumour doublings. $R$ subscript indicates values rescaled to population size of $10^{10}$. The parameters $f_{sub}$, $w$, and $t_s$ are all inferred. We approximate the rescaled subclone emergence time $t_{s_R}$ as

$t_s * log\left(N_{end_R}\right)/log(N_{end})$.

## Supporting information

**S1 Text. Pseudo-algorithms for generating synthetic tumours subject to positive selection or neutral evolution.**
(PDF)

**S1 Fig. An interactive figure of an example VAF distribution from a neutral synthetic tumour sequenced at varying depths.**
(ZIP)

**S2 Fig. An animated sample of paired simulations with positive selection (left) and neutral evolution (right).** Orange lines in the positive selection plot (left) indicate the subclone frequency (cellular proportion / 2).
(GIF)

**S3 Fig. Evaluating validity of synthetic data generation scheme with respect to real patient data (removal of low frequency variants based on mean sequencing depth).** To examine the specification of the synthetic data generation process, we iterated the VAF distribution feature vectors from each patient through the entire simulated training data set of 40 million synthetic tumours, searching for the nearest neighbours based on euclidean distance. We provide overlaid histograms of each patient VAF distribution with the closest nearest neighbour. All samples can be examined by using the dropdown menu. The feature vectors for this analysis were generated by using the mean sequencing depth to remove low frequency variants as described in Methods.
(ZIP)

**S4 Fig. Evaluating validity of synthetic data generation scheme with respect to real patient data (removal of low frequency variants based on mean effective coverage).** To examine the specification of the synthetic data generation process, we iterated the VAF distribution feature vectors from each patient through the entire simulated training data set of 40 million synthetic tumours, searching for the nearest neighbours based on euclidean distance. We provide overlaid histograms of each patient VAF distribution with the closest nearest neighbour. All samples can be examined by using the dropdown menu. The feature vectors for this analysis were generated by using the mean effective coverage (mean sequencing depth $*$ purity) to remove low frequency variants as described in Methods.
(ZIP)

**S5 Fig. Comparison of nearest neighbour search when using mean effective coverage versus mean sequencing depth.** Using mean effective coverage (mean sequencing depth $*$ purity) does not improve the fit between synthetic VAF distributions and patient tumour biopsy VAF distributions as defined by the euclidean distance of the nearest neighbour. Furthermore, samples only show modest increase in euclidean distance with decreasing purity which suggests the synthetic data generation scheme is robust to modest reductions in purity (e.g., $> 0.5$ analyzed in this study).
(PNG)

**S6 Fig. Accurately detecting positive selection and subclonality is dependent on the sequencing depth, number of subclonal mutations, and subclone frequency at time of biopsy.** For each VAF distribution, we computed the mean probability estimate across 25 stochastic passes through our trained neural network for the **(Top row)** both evolutionary mode classification, P(Selection), and the **(Bottom row)** number of subclone classification, P(N subclones). The interactive plots below show the mean probability estimates for both tasks at increasing subclone frequency (x-axis) and increasing subclone mutations (y-axis) for the top 25 trained deep learning models (dropdown menu). Hovering your cursor will show the mean probability estimate (z) at the given mutation-frequency combination. For P(Selection, we also provide the upper and lower bound of the 89% equal-tailed interval. In practice, to mitigate model overconfidence, we only call positive selection if the lower bound of the 89% interval is greater than 0.5. For reference, we use model TASYG7N3IJR1DLN in downstream inference tasks.
(ZIP)

**S7 Fig. Comparison of classical population genetic statistics, cancer evolution statistics from Williams et al. 2016 [12], and deep learning models for differentiating between positive selection and neutral evolution.** To evaluate each method, we simulated approximately 2.8 million synthetic tumours across different mean sequencing depths (50, 75, 100, 125, 150, 200, 250x) and sequencing overdispersion parameters (0, 0.001, 0.003, 0.01, 0.03). Sequencing

overdispersion refers to the rho parameter of the beta distribution in a beta-binomial sequencing noise model. An overdispersion parameter of 0.01 is a rough upper limit for empirical data. The mutation rate was set to 100 mutations per cell division per genome. **(a)** ROC curves for each sequencing depth and dispersion combination where columns indicate sequencing dispersion (see top text) and rows indicate sequencing depth (see right text). For each non-TumE statistic, the optimal threshold/cutoff value for specifying positive selection was determined by maximizing the sensitivity and specificity in a cutpoint analysis using the cutpointr package. **(b)** The median AUROC across all sequencing depth and overdispersion parameters for each method. TumE—Mean refers to calling positive selection if the mean of the approximate posterior density was greater than 0.5 for P(Selection). TumE—L89% refers to calling positive selection if the lower bound of an 89% equal-tailed interval was greater than 0.5 for P(Selection). **(c)** The precision and recall for calling positive selection using different approximate posterior cutoffs (mean or lower bound of 89% equal-tailed interval) at increasing sequencing depth. (PNG)

**S8 Fig. Predicting the number of subclones (0, 1, 2) in 2.8 million synthetic tumours.** To evaluate each method, we simulated approximately 2.8 million synthetic tumours across different mean sequencing depths (50, 75, 100, 125, 150, 200, 250x) and sequencing overdispersion parameters (0, 0.001, 0.003, 0.01, 0.03). We computed precision and recall across for predicting 0, 1, or 2 subclones. **(a)** Precision and **(b)** recall for prediction of 0 subclones. **(c)** Precision and **(d)** recall for prediction of 1 subclone. **(e)** Precision and **(f)** recall for prediction of 2 subclones. MOBSTER refers to the population genetics mixture model developed in [16]. (PNG)

**S9 Fig. Correlation between true subclone frequency and predicted subclone frequency using synthetic supervised learning (TumE) and a population genetics informed mixture model (MOBSTER).** We simulated approximately 1 million synthetic tumours with one subclone between 9–41% VAF across a range of sequencing depths (50 – 250x) and overdispersion parameters rho (0–0.03). To enable a reasonable comparison against the mixture model approach we only evaluated correlations where the predicted number of subclones for a given method matched the true number of subclones. We find strong agreement between both methods at higher sequencing depths. However, we note that the mixture model approach took approximately 24 hours using 500 cores to analyze >1 million synthetic tumours. Conversely, the trained neural network took a similar amount of time while using only one core. (PNG)

**S10 Fig. Error in predicting frequency of 2 detectable subclones with synthetic supervised learning (TumE).** We simulated approximately 500,000 synthetic tumours with 2 subclones located between 9–41% VAF. Our simulations covered sequencing depths from (50 – 250x) and sequencing overdispersion parameters (0–0.03). Each panel shows the predicted error (true frequency—predicted frequency) for the highest frequency subclone (Subclone 1) and lowest frequency subclone (Subclone 2) in each of the 500,000 VAF distributions/synthetic tumours. The mean percentage error (MPE) for predicting Subclone 1 and 2 is provided at each sequencing depth and overdispersion combination. The color of each point is the L1 loss or mean absolute error for each sample. (PNG)

**S11 Fig. Relationship between frequencies of subclones in the 2 subclone setting and the mean percentage error for the highest frequency subclone (1st subclone).** We simulated approximately 500,000 synthetic tumours with 2 subclones located between 9–41% VAF. Our simulations covered sequencing depths from (50 – 250x) and sequencing overdispersion

parameters (0–0.03). For frequency intervals/bins of 0.1 VAF, the mean percentage error was computed for predicting the 1st subclone frequency.
(PNG)

**S12 Fig. Relationship between frequencies of subclones in the 2 subclone setting and the mean percentage error for the lowest frequency subclone (2nd subclone).** We simulated approximately 500,000 synthetic tumours with 2 subclones located between 9–41% VAF. Our simulations covered sequencing depths from (50 – 250x) and sequencing overdispersion parameters (0–0.03). For frequency intervals/bins of 0.1 VAF, the mean percentage error was computed for predicting the 2nd subclone frequency.
(PNG)

**S13 Fig. Evaluation of TumE evolutionary classification estimates in an orthogonally simulated dataset of 900 synthetic tumours from Caravagna et al. 2020 [16].** (Top) VAF distribution for a given simulation. **(Bottom left)** Approximate posterior for P(Selection). **(Bottom right)** Approximate posterior for P(N subclones). Each simulation can be browsed in the dropdown menu.
(ZIP)

**S14 Fig. Subclone frequency estimates are only accurate at detectable frequency ranges (~10–45% VAF). (Left)** The distribution of predicted subclone frequencies when one subclone was correctly predicted (true) vs incorrectly predicted (false). **(Right)** True subclone frequency (as defined by TEMULATOR) versus predicted subclone frequency. Note that a subclone must appear between ~10–40% VAF for accurate estimation. Blue dots indicate correct prediction of selection. Red dots indicate incorrect prediction of neutrality.
(ZIP)

**S15 Fig. TumE performance (precision and recall) for predicting the number of subclones across 26 different birth rate and death rate combinations in 6.7 million synthetic tumours.** We evaluated model performance under different birth rate and death rate combinations as we built our inference models using data generated at a birth rate of log(2) and death rate of 0. We find that our performance is consistent across all birth rate and death rate combinations analyzed, although we are more likely to overestimate two subclones when only one subclone is present at low birth rate and death rate combinations (e.g. birth rate = 0.4 and death rate = 0.3).
(PNG)

**S16 Fig. TumE performance (precision and recall) for predicting the frequency of a single subclone across 26 different birth rate and death rate combinations in 6.7 million synthetic tumours.** We evaluated model performance under different birth rate and death rate combinations as we built our inference models using data generated at a birth rate of log(2) and death rate of 0. We find that our performance is consistent across all birth rate and death rate combinations analyzed. Correlation between true and predicted subclone frequency provided in top right corner of each facet.
(PNG)

**S17 Fig. Evaluation of the peak-finding/heuristic VAF adjustment method.** We evaluated TumE estimates in 6000 tumours with and without correcting for tumour purity. Tumour purity ranged from 0.2–1 and sequencing depth ranged from 20–200. Our analyses suggest tumour purity correction taking into account clonal peak adjustment and diploid purity adjustment (i.e. VAF/purity) is necessary to ensure accurate prediction in low purity samples.
(PNG)

**S18 Fig. False positive rate for positive selection ($>$ = 1 subclone) at variable sequencing depths, tumour purities, and errors in purity estimates in 6000 synthetic tumours.** We evaluated TumE estimates in 6000 synthetic tumours following the addition of different levels of incorrect purity estimates ranging from 1 to 25% above or below the true purity. VAFs from each synthetic tumour were corrected using the peak-finding/heuristic adjustment method described in Methods. We found TumE estimates using this adjustment were robust to errors in purity estimates up to 25%, maintaining a <5% false positive rate at mean effective sequencing depths of ~40x. The top labels on each panel column is the true tumour purity. The right labels on each panel row are the mean sequencing depth for each synthetic tumour. (PNG)

**S19 Fig. VAF distributions with annotated TumE fits for 75 PCAWG samples with either zero or one detected subclone.** Each tumour was classified as either neutrally evolving or subject to positive selection using TumE run with 250 Monte Carlo dropout samples. Samples subject to positive selection were further annotated as 1 or 2 subclone. This figure displays all TumE fits in samples classified as neutrally evolving or carrying 1 subclone. If a subclone was present, we annotated each plot with known driver mutations and the functional impact of the corresponding mutation. The statistics generated for the three samples identified with 2 subclones can be found in the S2 Table associated with the paper. (PNG)

**S20 Fig. Viable fitness and emergence time parameter combinations for detectable subclones (~10–40% VAF) in the TEMULATOR simulation framework.** Before generating a complete synthetic dataset, we examined the subclone fitness and emergence time that led to detectable subclone frequencings using the TEMULATOR simulation framework. Synthetic tumours were initialized with 500 clonal mutations and then simulated at a birth, death, and mutation rate of 1, 0.2, and 20 to a final population size of 10,000 cells. Mean sequencing depth was set to 100x and sequencing noise was generated under a beta-binomial model. **(a)** Subclone frequencies at varying fitness and emergence time combinations in the 1 subclone setting. **(b, c)** Subclone frequencies at varying fitness and emergence time combinations in the 2 subclone setting where the latest arising subclone is **(b)** and the earliest arising subclone is (**c**). We note that we only consider the 1 subclone case when validating the transfer learning approach in this paper. (PNG)

**S21 Fig. Comparison of predictive performance for inferring evolutionary parameters with and without pre-trained TumE models.** To exhibit the benefit of using pre-trained models when performing related evolutionary inference tasks, we evaluated the mean percentage error for predicting four TEMULATOR parameters/variables in a test set of 400,000 synthetic tumours. Non-pretrained models (False) and models with pre-trained convolutional layers (True) were trained on 500,000 synthetic TEMULATOR tumours for a maximum of 10 epochs. Hyperparameters (number of fully connected layers in new task specific branches and learning rate) were tuned via random search. Predictions for each model (38 non-pretrained & 33 pre-trained) were made across 50 sequencing depth and mutation rate combinations (e.g. 100 depth and 50–100 mutations per genome per division; n = 1900 & n = 1650). (PNG)

**S22 Fig. Evaluating the relationship between training set size and neural network model performance in a holdout test set of 500,000 synthetic tumours. (top row)** (Top row) Distribution of mean absolute percentage error (MAPE) versus training set size with 100 models

generated via random hyperparameter search per training set size. Each model was evaluated in 500,000 synthetic tumours simulated at mean sequencing depths of 75–200 and mutation rates ranging from 0–500 mutations / genome / division. Only synthetic tumours containing a subclone between 20–80% subclone cellular fraction were included. (**Bottom row**) Evaluating MAPE in the top model (based on minimizing MAPE) from each training set size. For each model, MAPE was computed across 300 subclone cellular fraction (0.2–0.8, step 0.1), mutation rate (0–500, step 50), and sequencing depth (75, 100, 125, 150, and 200) combinations. (PNG)

**S23 Fig. Comparison of mean percentage error with and without post-hoc mutation rate correction. (a)** We observed a modest but systematic underestimate of mutation rate following model training (as obseved in the positive mean percentage error). To correct this underestimate, we fit the outputted mutation rate estimates with a polynomial regression (degree 2) to predict the true mutation rate in a training set of 1000 synthetic tumours. **(b)** We then validated the correction in a holdout set of 100,000 synthetic tumours. The correction properly adjusted the majority of mutation rate estimates although lower mutation rates remained reasonable but underestimated. The labels on right axis of each plot indicate mean sequencing depth.
(PNG)

**S24 Fig. Mean percentage error for inferring parameters from TEMULATOR simulations (mutation rate, subclone cellular fraction, subclone emergence time, and subclone fitness).** Mean percentage error for predicting mutation rate (per genome per division), subclone cellular fraction, subclone fitness $(1 + s)$, and subclone emergence time (in tumour doublings) with top performing TumE transfer learning model. The labels on the right axis of each plot indicate mean sequencing depth.
(PNG)

**S1 Table. CanEvolve simulation parameters.** Simulation parameters used in the primary development and benchmarking of TumE models. The simulator used to generate this synthetic data can be found at https://github.com/tomouellette/CanEvolve.jl. And julia scripts for generating synthetic data (synthetic data D—F) can be found at https://github.com/tomouellette/TumE/bin/synthetic.jl.
(XLSX)

**S2 Table. TumE empirical estimates.** TumE estimates for the evolutionary mode, number of subclones, and subclone frequency in 88 whole-genome or whole-exome sequenced samples. Note that for completeness, we provide quantitative estimates for ALL parameters across all models. A python package for TumE can be found at https://github.com/tomouellette/TumE. Note that all PCAWG samples have ICGC specimen ids—these can be linked to aliquot ids via publicly available information on the ICGC data repository.
(XLSX)

**S3 Table. MOBSTER empirical estimates.** MOBSTER estimates in 78 PCAWG samples. MOBSTER was run with both ICL and reICL model selection. Default settings were used as outlined in MOBSTER package. VAFs were filtered above 5% VAF and below 70% VAF before application of MOBSTER. Only diploid regions were used. The number of subclones were determined as follows. Any sample contained in the set ({tail = TRUE, c1freq > 0, c2freq > 0}, {tail = FALSE, c1freq & c2freq & c3freq > 0}) were labeled 1 subclone, {tail = TRUE, c1freq & c2freq & c3freq > 0} were labeled as 2 subclones. The remaining samples were labeled 0 subclones.
(XLSX)

**S4 Table. SciClone empirical estimates.** sciClone estimates in 78 PCAWG samples. Default settings as outlined in sciClone package were used. VAFs were filtered above 5% VAF and below 100% VAF before application of sciClone. Only diploid regions were used. The estimated number of subclones was taken as the number of clusters detected by sciClone minus 2 (1 for clonal peak and 1 for neutral tail). If a sample was labeled -1 (e.g. only a clonal peak) it was called as 0 subclones.
(XLSX)

**S5 Table. TEMULATOR simulation parameters.** Simulation parameters used for generating additional synthetic data using an alternative simulation framework.
(XLSX)

## Author Contributions

**Conceptualization:** Tom W. Ouellette, Philip Awadalla.

**Formal analysis:** Tom W. Ouellette.

**Funding acquisition:** Philip Awadalla.

**Investigation:** Tom W. Ouellette.

**Methodology:** Tom W. Ouellette.

**Resources:** Tom W. Ouellette, Philip Awadalla.

**Software:** Tom W. Ouellette.

**Supervision:** Philip Awadalla.

**Visualization:** Tom W. Ouellette.

**Writing – original draft:** Tom W. Ouellette, Philip Awadalla.

**Writing – review & editing:** Tom W. Ouellette, Philip Awadalla.

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
