## [Decision Letter · Decision Letter 0]

3 Jan 2022

Dear Mr Ouellette,

Thank you very much for submitting your manuscript "Inferring ongoing cancer evolution from single tumour biopsies using synthetic supervised learning" for consideration at PLOS Computational Biology.

As with all papers reviewed by the journal, your manuscript was reviewed by members of the editorial board and by several independent reviewers. In light of the reviews (below this email), we would like to invite the resubmission of a significantly-revised version that takes into account the reviewers' comments.

We agree with the reviewers' comments below and encourage you to address their concerns, especially regarding noise in the simulations and comparisons.

We cannot make any decision about publication until we have seen the revised manuscript and your response to the reviewers' comments. Your revised manuscript is also likely to be sent to reviewers for further evaluation.

Sincerely,

Niko Beerenwinkel, Ph.D.

Guest Editor

PLOS Computational Biology

Jian Ma

Deputy Editor

PLOS Computational Biology

We agree with the reviewers' comments below and encourage the authors to address their concerns, especially regarding noise in the simulations and comparisons.

Reviewer's Responses to Questions

**Comments to the Authors:**

**Reviewer #1: **The authors have provided a method for generating synthetic bulk tumor sequencing data by modifying an existing algorithm. They also designed and implemented a novel deep learning model and train it on the synthetic data to capture tumor evolution, i.e. detect positive selection, identify number of subclones, and estimate subclone frequencies.

The data synthesis method and the deep learning model are well-designed and the authors show that the model can learn the patterns of the synthetic data well under different testing scenarios. They also show that the CNN embedding can be used for transfer learning. However, simulation-based machine learning models are only as good as the assumptions underlying the data synthesis process. In other words, only realistic assumptions during the data synthesis can result in trained models that are applicable to real-world data.

To that point, I believe the authors have made a fair amount of effort to show the validity of the simulated data. First, they use a well-established data simulation framework, i.e. stochastic branching, as the basis of their suggestions. Second, the authors evaluate their model on orthogonal synthetic data to showcase its generalizability. Third, it is shown in the supplementary figures 3 and 4 that some of the synthetic data are similar to real-world data from patient tumors. Fourth, the model has been able to perform well on 'gold standard' real data.

Given the computational accuracy and speed of the proposed solution and the novelty of the methods, I recommend this manuscript for publication.

**Reviewer #2: **This paper discusses the use of deep learning and transfer learning, coupled with Bayesian inference, to infer the evolutionary history parameters relevant to whole-genome or whole-exome sequencing of tumour samples. It is well-written and coherently argued, but there are several limitations that I would like to see addressed before the article can be published.

1) Although the software tool is easily downloaded and installed, the data used in the provided example (which comes from a public AML dataset) is not easily available. It would be imperative to ensure reproducibility by either providing a complete pre-processing procedure for this dataset or providing an already pre-processed version in the online supplements if that is an option.

2) This paper discusses transfer learning, but only in one direction (from other simulators to TEMULATOR), which begs the question of whether the transfer could be effective in the other direction (say, from TEMULATOR to other simulators). Also, although the amount of data used for retraining is relatively small (less than a million samples vs. tens of millions of samples), it is large in absolute terms, and it would be interesting to see a discussion of how the accuracy of the inference changes with the size of the retraining sample.

3) It was not clear to me why a paper focused on the evolutionary histories of tumours does not include a discussion of – or even reference to – any of the literature using phylogenetic methods (such as “SPhyR: Tumor Phylogeny Estimation from Single-Cell Sequencing Data under Loss and Error” and related one). I would like to either see a direct comparison to those methods or a complete explanation of why such a comparison is not appropriate, at least.

**Reviewer #3: **This paper presents TumE, a Deep Learning approach to tumour subclonal deconvolution. The main idea is to train a deep classifier to predict K - the number of tumour subclones - from synthetic tumour evolution datasets generated through a branching process model. The data-generating algorithms are taken from previous works by Williams et al., and comparison is carried out against 2 mixture-based methods (Sciclone and MOBSTER), with only one (MOBSTER) that accounts for neutral tails in tumour evolution.

I praise the following aspects of this work:

- reproducibility: almost all code is present and quite well documented. While this might seem obvious to a computational scientist, this is not the standard attitude in the field.

- Deep Learning: while we are not huge DL pratictioners, we found the idea of training over simulations the most interesting aspect of this paper.

The DL model - once trained - is obvsiously faster than mixture models, even all models are very fast, and TumE seems to be more precise in some scenarios. Clearly TumE has limitations in terms of interpretability and other features that are instead clear in mixture models (mutation assignents, and flexibiulity to scale to any subclonal architecture). Overall, one aspect of the TumE model that should be considererd is the adherence of the training set to real data, especially when the tools is applied to real datasets (in this sense some extra work has to be carried out). To improve the work we discusses major points below.

Major points (computational aspects for the training set):

- Overall, the method provides an interesting approach to the problem of reconstructing the subclonal structure of a single tumor sample, taking it from an unsupervised to a supervised learning setting and allowing for greater functional complexity than models that rely on mixtures. However, the network was trained on a set, which although huge, is composed only of simulated data. Ultimately, the quality of the tool will depend on how well these simulations are able to generalize to the real data. Despite the effort in generating the simulations we believe that some naive assumptions are made regarding how really "complex" is patient genomics data (heavily harbouring somatic alterations, with noise from upstream bioinformatics analysis, noise from specific sequencing technologies etc.). We therefore think it is important to consider also:

- The effect of miscalled CNVs segments in the tumour profile. The effect of such segments is to re-shape the VAF distribution, generating false VAF peaks (confounders). It would be nice if TumE could predict K - number of clones - but also propose "alternative" explanations such as "miscalled CNVs" (here, non-diploid segments called as heterozygous diploud). In this ways, a tumour would be "polyclonal" unless miscalled CNVs are a possible alternative explanation of data peaks (the user might afterwards screen off one model versus the other with complementary analyses). This will be crucial to make sense of TumE's predictions on real data (see below). Can such error probability be included in the learning process? Can TumE decide whether data has strong-enough signals to support one model over the other in an undisputable way?

- After the MOBSTER paper, the quest of whethere a dataset harbours resolution for reliable subclonal deconvolution seems important. For instance, one conclusion of that paper is that PCAWG data are too-low resolution to trust most deconvolutions

-"Our analysis suggests that, for most PCAWG cases, the data resolution was too low to conduct robust subclonal reconstruction. [...] Standard analyses of these data therefore risk systematically mistaking neutral tails for subclonal clusters, thus inflating the complexity of the inferred subclonal architectures and producing incorrect phylogenetic trees.".

I interpreted that as a cautionary message - garbage in, garbage out - which I think it is reasonable because, at low-coverage/purity, certain noise sources (callers?) are difificult to control. Ouellette and Awadalla should consider if they can reliably learn to regress low-resolution data assuming - as they do now - almost a noise-free model (see other points below), or at least a model where the noise contribution is very limited to be considered "realistic".

- The effect of miscalled purity. While the authors shift randomly the clonal per between 45% and 50%, we believe a more detailed approach is needed (changes in purity also shift sub clonal clusters and tails). In particular, the estimation of sample purity happens together with CNV inference - i.e., the two quantities are linked. For this reason, miscalling purity often implies miscalling ploidy, and therefore miscalling clonal heterozygous diploid segments (allele-specific CNVs). The effect of these chained errors is not considered in the current problem formulation, and the consequent learning process is working in too idealistic scenarios. The authors should either try to augment their model and create more realistic datasets to train TumE, or at least have a more critical undertaking of these aspects.

- While randomly trimming the tail partially reproduces problems of noisy mutations at low frequency, in reality what happens is more an undersampling process, where we do see some mutations at low frequency, just way fewer than expected. Also, there is an intrinsic noise effect that is due to the particular caller adopted (certain callers adopt different calling thresholds, noise models etc.), which is also exacerbated in the case of consensus pipelines like in PCAWG, where multiple callers are adopted (what is the consensus noise model!?). Randomly chopping the tail is not necessarily neither the best nor the correct model - these aspects should be clarified. As a matter of the experience of this reviewer in working iwth noisy low-frequency data, whereas this shhold be perfectly power-law shaped (theoretically), it often shows exrta peaks after the effect of any caller is added up. This effect often resembles a sort of non-linear noise model because of heuristics adopted by callers (i.e., retain mutations with at least 3 supporting reads, etc.). The authors should therefore provide more evidence about the goodness and appropriateness of their random-truncating tail approximation.

- Mutation assignments: a foundamental feature of mixture models is the explicit modelling of a latent responsibility per mutation. TumE regresses K over the VAF, producing a multinomial estimate of the number of subclones, and just afterwards takes care of assigning mutations to clusters. The process of clustering assignements is performed by using the correct model, but still it is completely decoupled from the regression. The authors should comment more on this and the potential pitfalls - in the end, the authors claim that they can learn complex non-linear VAF/data relations, but finally cannot perform clustering assignments with anything than a standard approach, therefore losing potential advantages of the non-linear mapping. In this sense, mixtures are ahead of TumE. Also, given the presence of ground truth data where assignments are known, quantitative measures of performance on miscalled assignments must be provided. For this purpose, mutual information and related measures should be used to provide a quantitative measure of how TumE assignements are off, also relative to other models adopted (SciClone and MOBSTER).

Major points (computational aspects for the training set):

- One aspect is not completely clear to us. As far as we understand, the TEMULATOR population genetics model by Heide et al is the same of the Julia model by Williams et al. The simulation model by the authors is also a re-implementation [*] of the model by Williams et al, with some modification that augment generality but do not describe completely different population genetics processes. Therefore, we do not understand if learning parameters of TEMULATOR after training on simulations à la Williams should be, or not, a great surprise. We would be more surprised if parameters of completely different models were learned by transfer learning - probably after suitable transformations. We are however certain about one thing: learning decoupled growth and death rates is very interesting - probably the most interesting thing, since all other parameters are analytical from VAF fits according to Williams et al. In general, a more thorough discussion about these simularities among the generative models would help this reviewer to appreciate that part of the paper.

[*] The main differences claimed regard the possibility of randomly sampling certain parameters, which are constant in the Williams/Heide models to speed up simulation. While this includes random effects that add learning overhead - i.e. the stochasticity of the probability of a mutation to be driver - it is not clear how different are the 3 proposed models (Heide, Williams, Ouellette).

Major points (real data applications, partially overlapping with the points regarding trainig data):

- In the analysis of 85 PCAWG samples the authors claim to detect positive selection in ~38% of cases, compared to ~96% in the original PCAWG analysis and ~3% in Caravagna et al. There is clearly a strong point of controvercy across all analyses, but I see 2 problems with these summary statistics:

- Samples set: it should be clarified if the pool of samples upon which these analyses are carried out is the same (clearly is not), otherwise comparing these summary statistics is pointless or requires, at least, some caution. The overall PCAWG paper analyses the full set of samples (~3000), the paper by Caravagna et al. focuses on a subset of samples with high-purity and high-coverage in order to determine samples that have high-enough signal quality to determine the site frequency spectrum of the tumour's neutral tail, plus subclones. This paper selected 85 cases - "spanning 8 different cancer types, retrieved from the pan-cancer analysis of whole genomes (PCAWG)" - where the rationale for sample inclusion seem to be explained in Supplementary Figure 18 (with constraints on sample purity and coverage), but this does not match up with the one in Caravagna et al, for instance. With this inconsistencies we cannot be confident about the meaning of the proposed comparisons, and therefore the reliability of conclusions. This aspect has to be clarified, and any statment about "subclonal selection" in real data needs to be weighted by careful evidence.

- Data suitability: here it comes an important limitation of having trained TuME on "perfect" data. The perfect training set of TumE is ideal compared to real data where somatic calling is likely imperfect. In practice, the authors seem to have taken for granted copy number estimates from PCAWG (and indeed adopted just a purity adjustment). Errors in CNA calling will create artificial bumps in the data distribution, which will "confound" VAF-based inference tasks, giving the illusion of subclones. Similarly, subclonal CNAs - as defined in PCAWG - might also affect the VAF disitribution (are they to be removed in your analysis?). With reference to Supplementary Figure S18, the authors should show, for each one of the 85 cases, the result of TumE analysis on the 85 samples, subclonal clustering and potential neutral mutations assignments. Heuristics should also be adopted to assess wethere the input data for TumE is consistent with the assumptions of the method itself. Otherwise, it is difficult to assess the quality of the inference of the proposed method on real data. Once this analysis has been properly carried out, how many subclonal drivers (at least, simple mutations) can the author assign to the identified 40% of subclonal expansions? Also, can the authors present the analysis with alternative models (MOBSTER, Sciclone) for these samples? Are they consistent and, if not, where do they differ?

Minor points:

- Difference in performance: TumE is far more fast than all the methods. The authors indeed report that (Supplementary Figure S9)

"[..] we note that the mixture model approach took approximately 24 hours using 500 cores to analyze >1 million synthetic tumours. Conversely, the trained neural network took a similar amount of time while using only one core."

However, we feel that all methods are quite fast, also considering that they provide more information than TumE concerning clustering assignments etc. We do actually feel that all tested methods are adequately fast, providing results in about <<10 minutes with real data samples. Also, we would like to note that not even at ICGC... there are millions of tumours to run. Historically, I guess one of the problems in the field was that, a few years ago, and certainly at the time of PCAWG, the speed of Monte Carlo samplers like those in DPClust, PyClone (before its Pyclone-VI extension), etc. were a real bottlneck. Nowadays, once that more powerful variational models (like in Sciclone) or MLE estiamtes (like in MOBSTER) , I am not sure "speed" is still a bottleneck; the authors could comment on this.

- Performance of alternative models: it seems that MOBSTER has been run fitting with more configurations then other methods. In particular, if we understand right it has been running selecting bewtween models with and without tails on simulated data, but in the end, you do not distinguish if the "tail" cluster is fit to a tail or not. Given the number of K-values tested (1,2,3 as of limits in TumE), isn't this testing twice the fits with this method? If yes, shouldn't you at least divide by 2 its computational time?

- It is generally true the DL can provide very complex non-linear representation of the input data, it would be interesting to analyse some cases where TumE performs better than competitors and trying to get what are the features that are actually improves the inference over a standard mixture models. The comparison would be helful also the other way round, showing when the mixture model provides more precise results than TumE. I am wondering if any explanability argument can be raised in these tests.

- In figure 4c how do you explain the error to deviate from normal (ex. 2 picks or skewed distribution)?

**Have the authors made all data and (if applicable) computational code underlying the findings in their manuscript fully available?**

Reviewer #1: Yes

Reviewer #2: Yes

Reviewer #3: Yes

PLOS authors have the option to publish the peer review history of their article (what does this mean?). If published, this will include your full peer review and any attached files.

Reviewer #1: No

Reviewer #2: No

Reviewer #3: No
---

## [Decision Letter · Decision Letter 1]

9 Mar 2022

Dear Mr. Ouellette,

We are pleased to inform you that your manuscript 'Inferring ongoing cancer evolution from single tumour biopsies using synthetic supervised learning' has been provisionally accepted for publication in PLOS Computational Biology.

Best regards,

Niko Beerenwinkel, Ph.D.

Guest Editor

PLOS Computational Biology

Jian Ma

Deputy Editor

PLOS Computational Biology

Reviewer's Responses to Questions

**Comments to the Authors:**

Reviewer #1: none

Reviewer #3: The author answered most of our questions thoroughly; in the end detecting selection from these data is difficult and will only be convincing once (experimental) validations will be possible.

The new additions in the text allow users to have a more complete understanding of how to prepare the input for the tool and of the possible sources of noise that may invalidate the analysis. While we consider the approach presented here a notable demonstration of the use of Deep Learning in the field of tumour evolution; there are still some aspects regarding the application to real dataset that are not fully clear and that the authors might consider clarifying:

- The NN analysis looks pretty convincing, and shows how real dataset have a close match in the simulation space. By looking at the fits of the selected PCAWG samples in Supplementary Figure 19, we note how some of them have very convincing clonal peaks, while other look like they have some sort of split in the clonal clusters (with the subclonal density almost specular to the right tail of the clonal cluster). It would be interesting to look at the closest simulations for those cases and see how those are classified (neutral or with subclones) and what are the parameters that generate such data.

- While it is true that one can decouple peak inference and clustering, we may have a big mismatch between the actual internal representation of the subclonal density and the one used to perform mutations assignment.

**Have the authors made all data and (if applicable) computational code underlying the findings in their manuscript fully available?**

Reviewer #1: Yes

Reviewer #3: Yes

PLOS authors have the option to publish the peer review history of their article (what does this mean?). If published, this will include your full peer review and any attached files.

Reviewer #1: No

Reviewer #3: No

---

## [Editor Report · Acceptance letter]

7 Apr 2022

PCOMPBIOL-D-21-02073R1 

Inferring ongoing cancer evolution from single tumour biopsies using synthetic supervised learning

Dear Dr Ouellette,

I am pleased to inform you that your manuscript has been formally accepted for publication in PLOS Computational Biology. Your manuscript is now with our production department and you will be notified of the publication date in due course.

With kind regards,

Livia Horvath
